# Scalable computation of anisotropic vibrations for large macromolecular assemblies

Jordy Homing Lam [1,2,3], Aiichiro Nakano [1,4,5] ✉ & Vsevolod Katritch [1,2,3,6] ✉

The Normal Mode Analysis (NMA) is a standard approach to elucidate the anisotropic vibrations of macromolecules at their folded states, where low-frequency collective motions can reveal rearrangements of domains and changes in the exposed surface of macromolecules. Recent advances in structural biology have enabled the resolution of megascale macromolecules with millions of atoms. However, the calculation of their vibrational modes remains elusive due to the prohibitive cost associated with constructing and diagonalizing the underlying eigenproblem and the current approaches to NMA are not readily adaptable for efficient parallel computing on graphic processing unit (GPU). Here, we present eigenproblem construction and diagonalization approach that implements level-structure bandwidth-reducing algorithms to transform the sparse computation in NMA to a globally-sparse-yet-locally-dense computation, allowing batched tensor products to be most efficiently executed on GPU. We map, optimize, and compare several low-complexity Krylov-subspace eigensolvers, supplemented by techniques such as Chebyshev filtering, sum decomposition, external explicit deflation and shift-and-inverse, to allow fast GPU-resident calculations. The method allows accurate calculation of the first 1000 vibrational modes of some largest structures in PDB (> 2.4 million atoms) at least 250 times faster than existing methods.

The Normal Mode Analysis (NMA) is a standard approach to derive motions from static snapshots of a macromolecular structure. As a computational probe to shape-changing motions, the analysis has found wide applicability in the refinement and fitting of macromolecular structures[1,2] and in the simulations of functional motions when coupled with enhanced sampling techniques[3,4]. Many of these predicted modes of motion align consistently with available experimental data in kinases[5], ion channels[6] and transporters[7]. Such successful applications have led to an increasing appreciation of the

interdependence among biological structure, dynamics, and function[8]. In NMA, the input macromolecular structure is assumed to be in a conformational minimum of its potential energy. Spatial arrangements of atoms (or atom groups like residues) and forces among them are then assimilated into the Hessian matrix and the equation of motion is analyzed under the harmonic approximation. The outcome of NMA are eigenvectors, representing sets of optimal displacements for each atom in the system, ranked by their ascending eigenvalues that reflect increasing strain along those directions. Early attempts of NMA on

[1]Department of Quantitative and Computational Biology, University of Southern California, Los Angeles, CA, USA. [2]Bridge Institute and Michelson Center for Convergent Biosciences, University of Southern California, Los Angeles, CA, USA. [3]Center for New Technologies in Drug Discovery and Development, University of Southern California, Los Angeles, CA, USA. [4]Department of Physics and Astronomy, University of Southern California, Los Angeles, CA, USA. [5]Department of Computer Science, University of Southern California, Los Angeles, CA, USA. [6]Department of Chemistry, University of Southern California, Los Angeles, CA, USA. ✉e-mail: anakano@usc.edu; katritch@usc.edu

macromolecules[9–12] were derived from all-atom potentials and full dense Hessian matrices. This approach to NMA is still the golden standard in analyzing the shear and hinge motions of proteins with sizes typically less than 5000 atoms[13]. However, applying NMA in this form to larger macromolecules is difficult as their energy minimizations are prone to overstepping, rendering unphysical modes[14,15]. In a pioneering work by Tirion[15], this requirement was relaxed by replacing the detailed all-atom potentials with an elastic network of pairwise Hookean potential, connecting atoms within a neighborhood boundary. The NMA analysis then proceeds by assuming that the provided structure was minimized under experimental conditions. As such, crystal coordinates, which lack hydrogens and/or sidechains, can be analyzed directly in absence of a forcefield. This practical form of NMA is now commonly known as the elastic network model (ENM). Despite its simplicity, as demonstrated convincingly by Hinsen[16,17] and Sanejouand[18], the low-frequency ENM modes were shown to agree with the fluctuations observed in X-ray crystallography as well as with NMAs performed under standard all-atom potentials, as long as only low-frequency modes are concerned. The ENM also found major applications in Cryogenic Electron Microscopy (Cryo-EM) and Cryogenic Electron Tomography (Cryo-ET) for the flexible refinement of the composite density maps[19–21] and the flexible fitting of atomic models[22–24] when EM volume or atomic templates are available. These applications of NMA are especially compelling for large macromolecular complexes, when other methods are computationally expensive. However, computing the NMA, even in this practical ENM form, remains challenging for large macromolecular systems of more than 1,000,000 particles. Namely, the storage, the construction and the diagonalization of the Hessian matrix all create imminent difficulties in terms of time and memory complexities as the size of the matrix increases quadratically with the number of atoms $N$. While ideally the memory complexity of NMA only increases linearly, the linear factors due to packing density of atoms as well as the cubic time complexity to diagonalize the Hessian matrix present a major bottleneck in calculations (See Methods). As such, many innovative approaches, including Anisotropic Network Model (ANM)[15,25], Rotation-Translation Block (RTB) method[26] and Block Normal Mode (BNM)[27], were undertaken to eliminate the degrees of freedom (DOF) to be differentiated in these larger systems, hence reducing $N$ for 8-10 times. Agreement of these methods with experimental data[18,25] are satisfactory, though the displacement information of the discounted atoms were lost and the appropriate level of granularity, especially in presence of elongated (e.g., lipids) or planar (e.g., aromatics) chemical moieties, are hard to be determined. To apply NMA without excessive coarse-graining, the development of faster numerical recipes is necessary[28,29]. More recently, the choices on diagonalization algorithms were examined on a hyperthreaded machine in the work of Koehl[30]. This ultimately allowed the first 100 ENM modes of a ZIKV virus (PDBID: 5IZ7), with around 800 thousand atoms and around 300 million nonzero entries in its Hessian, to be calculated within an hour when the Jacobi-Davidson Method (JDM)[31,32] were filtered with an 80-degree low-pass Chebyshev polynomial of the Hessian[30]. However, the current state-of-the-art is still far from handling megascale systems with more than a million atoms and billions of non-zero entries in the Hessian.

In this work, we designed several synergistic algorithms to fully engage dense graphic processing unit (GPU) kernels in both construction and diagonalization of the Hessian in atomic NMAs. Specifically, we developed a level-structure algorithm to minimize the bandwidth of the Hessian, prior to its construction, and thereby converting its sparse computation into a globally-sparse-yet-locally-dense computation, which enables the batched execution of tensor products on GPUs. We also mapped, optimized and compared several low-complexity Krylov-subspace eigensolvers, supplemented by techniques such as Chebyshev filtering[33–35], sum decomposition, external explicit deflation[36] and shift-and-inverse, to allow fast calculations on a GPU device. The level-structure algorithm produces an isomorph of the original elastic network, for which its unpermuted eigenpairs can be recovered in linear complexity using the bijection mapping. The implementation of this INCHING ("Isomorphic Nma Calculations Harnessing 1 Necessary Gpu") algorithm presented here was benchmarked on macromolecules from the Protein Data Bank[37] with sizes up to 2.4 million atoms. Compared to other existing methods, 250–370 times speedup in throughput was achieved while maintaining residual error under $10^{-12}$. The utility of the INCHING method was also demonstrated through examples, including the largest experimentally resolved atomistic structure of a mature HIV-1 capsid[38] (PDBID:3J3Q) with 2.4 million atoms and 1.6 billion non-zero entries in its Hessian. Using our INCHING program, we were able to resolve its first 64 atomic normal modes within 44 min of wall-clock time on a single NVIDIA® A100 Tensor Core GPU and its first 1000 atomic normal modes within 63 h. Fast, accurate, and highly scalable GPU-optimized implementation of NMA approach will find practical applications in conformational analysis of large and dynamic macromolecular complexes, and facilitate their refinement from cryo-EM/cryo-ET data.

## Results

### Overview of the INCHING algorithm

The NMA is a study of a high-dimensional potential energy surface under harmonic approximation. In this work, the INCHING algorithm is applied to the elastic network model (ENM), a practical form of NMA introduced by Tirion[15]. As described in Methods, NMA is based on solving a standard eigenproblem, $\mathbf{HQ} = \mathbf{Q\Omega}$, concerning a Hessian matrix $\mathbf{H} \in \mathbb{R}^{3N \times 3N}$, where $\mathbf{\Omega} \in \mathbb{R}^{3N \times 3N}$ and $\mathbf{Q} \in \mathbb{R}^{3N \times 3N}$ are the eigenvalue matrix and the eigenvector matrix, respectively. For large collective conformational changes, only the eigenpairs with the smallest non-zero eigenvalues (the lowest-frequency modes) are of interest. However, challenges in the construction and diagonalization of the Hessian grow with the system size, and are notoriously resistant to parallelization. Consequently, current approaches to NMA are not readily adaptable for GPU computing, which imposes even stricter memory limits and algorithmic requirements to fully leverage on-chip parallelism. As illustrated in Fig. 1a, the INCHING algorithm was implemented in three synergistic stages—permutation, construction, and diagonalization—to resolve the challenges of calculating NMA on GPU. In the following sections, we will describe each stage in detail.

### Permutations to achieve bandwidth reduction

GPUs are hardware engineered to efficiently access contiguous memory locations, thus inherently favoring dense computations. However, in practice, the experience of memory access is dictated by the sparsity pattern of the data presented. Since the inception of NMA, it has been recognized that the Hessian is globally-sparse with prevailing zeros, primarily due to the long-range cut-offs applied to molecular mechanics potentials[11]. These globally-sparse Hessians are also locally-sparse, requiring very large bandwidths with non-uniform adjacencies (Fig. 1b–f, c.) due to sequentially distal segments of polymer(s) interacting in tertiary and/or quaternary structure(s)[39,40] (Supplementary Fig. 1), and the situation is further complicated by chemicals (e.g., cofactors, ions, water, lipids) with no inherent order being integrated into the polymer structure (Supplementary Fig. 2). All these factors contribute to degrade on-chip parallelism when dense row-sweeps were performed in batches.

The primary objective of the INCHING algorithm is to strategically permute columns and rows of the Hessian, prior to all its computations, such that a globally-sparse-yet-locally-dense computation can be achieved in subsequent stages. By permuting the atom ordering, we can always generate a graph isomorph of the original elastic network, while allowing retrieval of its unpermuted eigenpairs in linear complexity if a bijection mapping is provided. However, finding a permutation that exactly minimizes the bandwidth of a matrix is NP-

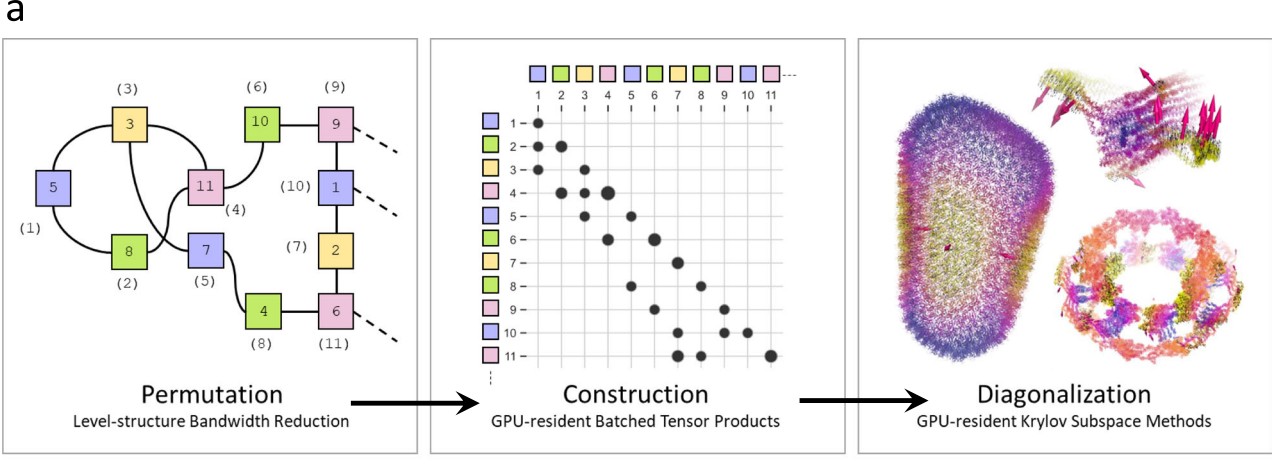

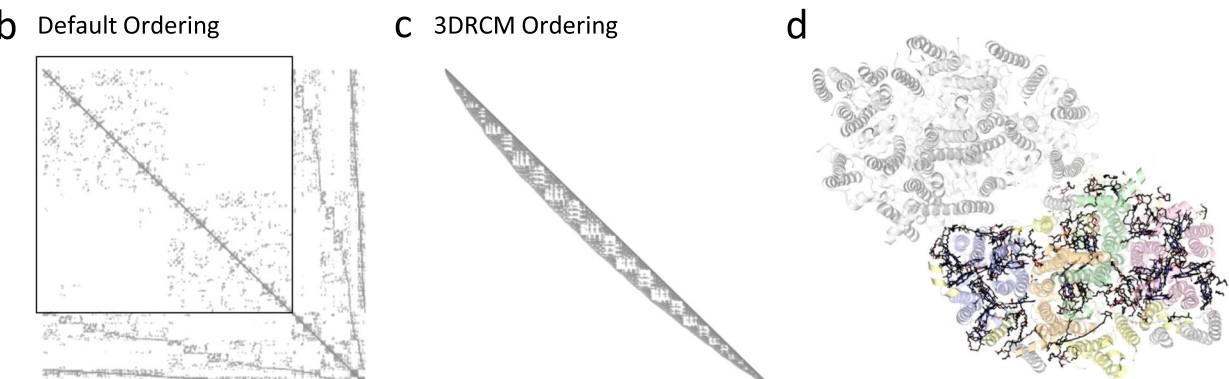

**Fig. 1 | Overview of the INCHING algorithm. a** Our Isomorphic NMA Calculations Harnessing 1 Necessary Gpu algorithm (INCHING) consists of three stages, namely, permutation, construction, and diagonalization. In the permutation stage, the 3-Dimensional Reverse Cuthill-McKee algorithm (3DRCM) takes atom coordinates as the input and produce a bandwidth-reduced indexing as output. The initial atom ordering is denoted within square-shaped nodes, while the 3DRCM-permuted atom ordering is represented using parentheses. In the construction stage, the lower triangle of a Hessian with globally-sparse-yet-locally-dense pattern is constructed incrementally on a Graphic Processing Unit (GPU) with tensor products and broadcasts. In the diagonalization stage, the eigenproblem is solved on GPU to produce depictable mode shapes. **b** Sparsity pattern of the Hessian with default atom ordering for the PsbM-deletion mutant of photosystem II. The top left square highlighted corresponds to the proteins in the macromolecular complex, which also comprises cofactors, lipids and water molecules. **c** Sparsity pattern of the Hessian lower triangle with 3DRCM permuted atom ordering for the same complex (**d**) overall spatial distribution of the chemicals intercalated in multiple protein chains in default sequential order (green, blue, pink, orange, yellow). Only one of the dimeric halves of the structure is colored, the rest of the macromolecule is shown as a gray surface; note that the complex is asymmetric due to slight difference in lipids and cofactors intercalated. The Hessian matrix in default ordering has a much longer bandwidth everywhere than that with the 3DRCM ordering.

complete[41]. To approximate this reduction, we have designed a level-structure algorithm called 3DRCM, which extends on the well-established Reverse Cuthill-McKee (RCM)[42,43] algorithm, to handle 3-D coordinate data. Traditional RCM and variants[44–46] operate under the assumption that the input —a matrix— is realized and stored once-and-for-all. This framework is suitable for testing new offline analyses on the preserved matrix. However, a major problem overlooked is that it may not even be practical to realize the matrix efficiently, when its parallel computation spans a large bandwidth. In 3DRCM, a 3-D-tree data structure[47] is taken as input instead of a matrix to eliminate this circular reference. This modification enables efficient dynamic neighborhood lookups in the permutation process and avoids premature realization of the Hessian matrix. In Supplementary Fig. 3a, we showed that for macromolecules with over 100 thousand atoms, the mean bandwidth for the Hessian in default PDB sequence ordering can vary between 10% and 90% of $N$. However, by permuting the Hessian with 3DRCM before its construction, the mean bandwidth of the Hessian is consistently reduced and remains below 10% of $N$, hence confirming the local density of the matrix. In Method and Supplementary Fig. 3b, we also showed that the time complexity of 3DRCM is practically dominated by a linear term proportional to the packing density of the macromolecule, thereby ensuring the cost-effectiveness of the algorithm.

## GPU-resident construction of the Hessian with batched tensor products

The proposed 3DRCM permutation enables spatial neighbors from each batch of atoms to be retrieved as locally-dense indexing slices. In our Method, we further showed that, by extending on the tensorial representation suggested by Koehl[30], the Hessian can be constructed in batches through vectorized operations, such as tensor products and broadcasts, to leverage efficient on-chip parallelism in GPU. This replaces the need for explicit index loading and external storage of the distance matrix to enable fast computations. Besides, in our implementation, only the lower triangle of the Hessian was incrementally stored, and later accessed, in Compressed Sparse Row (CSR) format, this halves the memory requirement. In Supplementary Fig. 4, we showed that even for a system comprising 2.4 million atoms (PDBID: 3J3Q), the computation of the 3DRCM permutation ordering and the subsequent construction of the Hessian on GPU took only 5.2 min and

6.9 min in wall-clock time, respectively. Our method shows significant speedup over ProDy2.4[48], an open-source implementation intended for coarse-grained protein representation in ANM tasks. For the largest case that ProDy2.4 handled (PDBID: 6NCL), containing 305 thousand atoms, ProDy2.4 took 7.8 h to construct the Hessian, whereas our INCHING approach delivered the matrix in just 93 seconds, showing a 302-fold acceleration.

## GPU-resident diagonalization of the Hessian matrix

In NMA calculations, a significant portion of computational resources is often dedicated to diagonalizing the Hessian, where the throughput is dependent on the performance of the eigensolver. One popular method in solving large symmetric eigenproblem is the Implicitly Restarted Lanczos Method (IRLM)[49], which is available in the ARPACK package[50], a backend incorporated into ProDy2.4. In ARPACK, the matrix-vector multiplications are accelerated by threaded BLAS subroutines. However, despite the utilization of a high-end AMD processor with 64 threads, ProDy2.4 routines (ProDy-ARPACK-FullSparse) fail to converge within 48 h when systems exceed 305 thousand atoms (PDBID: 6HIV, 311 thousand atoms), indicating that we have reached the limit of hardware improvement. Note that ProDy2.4 with the default LAPACK[51] backend (ProDy-LAPACK-FullDense) working on a full dense Hessian is faster than all other programs when there are less than 500 atoms, but we were not able to proceed once exceeding 29 thousand atoms due to rapid rise in memory consumption. We also implemented a version of INCHING (INCHING-ARPACK-FullSparse) that uses our fast Hessian construction routine on GPU followed by a one-time device-to-host transfer to carry out diagonalization with ARPACK, but we were not able to proceed beyond 1.2 million atoms in 48 h, reflecting the bottleneck in calculation is fundamentally the diagonalization process.

GPU-accelerated approaches for solving large sparse symmetric eigenproblems continues to be a vibrant area of research[52–57]. To explore this next-generation technology, we implemented, optimized and compared several diagonalization methods on a single NVIDIA® GPU device, including the Implicitly Restarted Lanczos Method (IRLM)[49,50], the Thick Restart Lanczos Method (TRLM)[58], the Jacobi Davidson Method (JDM)[31,32] and some of their Chebyshev-filtered versions e.g. the Chebyshev-filtered Thick Restart Lanczos Method (CTRLM)[34] and Chebyshev-Davidson Method (CDM)[35]. In general, these methods all share the objective of computing eigenpairs within an interval (e.g., those with the smallest eigenvalues), but they differ in their approaches to update the Krylov subspace that approximates the eigenpairs. (See Methods for detail). The IRLM and TRLM are variants of the Hermitian Lanczos Method (HLM), but they differ in how they utilize the solutions of a much smaller tridiagonal eigenproblem to reinitialize the Krylov subspace at restarts. On the other hand, the JDM directly corrects the Krylov subspace by solving for an approximate fit to eliminate residuals and subsequently also works over solutions of a smaller projected eigenproblem. The CTRLM and CDM uses filters (low- or band-pass) constructed from Chebyshev polynomials to magnify wanted interval in the spectrum. The choice among these methods is not obvious, though their speed all depends on the cost of matrix-vector multiplication. In our INCHING protocols (INCHING-TRLM-HalfSparse, INCHING-IRLM-HalfSparse, INCHING-JDM-HalfSparse, INCHING-CTRLM-HalfSparse, INCHING-CDM-HalfSparse), these multiplications are accelerated by the SpMV CSR kernel in cuSPARSE[59]. The GPU computation was instructed through CuPy[60], a python API to NVIDIA®'s CUDA[61], cuBLAS[62], cuSPARSE[63] and custom kernel programming. To accommodate the lower memory capacity of GPU, the Hessian is accessed as a sum decomposition of its lower triangle, halving the memory requirement. Alternatively, the calculations can also be done in the full sparse matrix with a further 40% speedup, when GPU memory is not exhausted. (See Supplementary Fig. 5) In Supplementary Fig. 6, we further showed that an explicit

external deflation[36] of the first 6 rigid modes with zero eigenvalues (rotations and translations) can help to improve runtime of the remaining 58 non-zero eigenpairs in sub-megascale regime while sharing very similar memory footprint. Finally, by incorporating low- and band-pass Chebyshev filters[34,57], we also lifted the memory limit regarding the number of modes to be resolved. (See INCHING-CTRLM-HalfSparse in Fig. 2 and Supplementary Figs. 7 and 8)

## Benchmarks

Benchmarks on correctness, memory and speed for calculating the first 64 modes are illustrated in Fig. 2. All reported runtimes are wall-clock time, and the tests were conducted on a computer with a 64-threads AMD EPYC™ 7513 processor and a single NVIDIA® A100 Tensor Core GPU, unless specifically noted. The accuracy is measured by the 2-norm of the residual error. Across 116 benchmark cases encompassing macromolecules ranging from around a thousand atoms to around 2.4 million atoms, including PDBID:3J3Q, the largest experimentally resolved atomic structure in PDB at <10Å range, our INCHING protocols were able to afford accuracy at $10^{-12}$ level and achieved a peak memory consumption consistently lower than the ProDy2.4 routines. Importantly, with memory requirement halved by accessing only the lower triangle of the Hessian with a sum decomposition, the throughputs of our INCHING protocols, including construction and diagonalization of the Hessian, are still 146–251 times faster than ProDy-ARPACK-FullSparse and are 265–1290 times faster than ProDy-LAPACK-FullDense, depending on our choice of diagonalization algorithm. Remarkably, for a 305-thousand-atoms system (PDBID: 6NCL), all of our INCHING protocols took at most 5 min to converge with the fastest convergence being achieved by INCHING-TRLM-HalfSparse at 2.9 min. The same task took ProDy-ARPACK-FullSparse 12.4 h to converge. INCHING-TRLM-HalfSparse, INCHING-JDM-HalfSparse and their Chebyshev-filtered versions (INCHING-CTRLM-HalfSparse, INCHING-CDM-HalfSparse) also converged for the HIV-1 capsid structure (PDBID:3J3Q, 2.4 million atoms), with the fastest convergence achieved by INCHING-CTRLM-HalfSparse within 44 min. In Supplementary Fig. 7, we show that, at the same accuracy level as JDM ($10^{-12}$), moderate speed-up in sub-megascale regime (mean at 1.21) can be achieved by CDM with an 80-degree polynomial, though the speed-up diminished to a slow-down in the megascale regime (mean at 0.91). This contrasts with applying an optimized low-pass filter to TRLM, where we showed that the Chebyshev-filtered TRLM (CTRLM) can steadily deliver a speed-up over TRLM in the megascale regime (mean at 7.44). In Supplementary Fig. 8, we further showed that linear or better scaling in runtime has been achieved in terms of the radii $R_C = 6,8,14$Å to be considered, except for several cases at a lower radius $R_C = 6$Å, likely due to poorer conditioning of the matrix. In Fig. 3, by incorporating a band-pass Chebyshev filter developed in recent works[34,57] into our TRLM implementation (INCHING-CTRLM-HalfSparse), we also achieved linear time scaling in the number modes to be solved, while keeping memory usage constant. This ultimately allows 1000 modes of all the benchmarks to be solved, in batches of 64 eigenvectors on a standard A100 NVIDIA® GPU, without compromising memory or run time or atomic details. The method was also tested on a much less expensive RTX4090 NVIDIA® GPU in batches of 28 eigenvectors in Supplementary Fig. 10.

## Anisotropic vibrations of some megascale systems

To illustrate the usage of our software, we have applied INCHING to some of the largest atomic objects available, both natural and artificial. In Fig. 4, we illustrate the first non-zero mode of the mature HIV-1 capsid structure (PDBID:3J3Q), the largest experimentally resolved atomic object to date at <10Å resolution range, containing 2.4 million atoms and 1.6 billion non-zero entries in its Hessian. The cone-shaped capsid is an assembly of 186 hexamers and 12 pentamers, and it was suggested that these pentamers located at its hemispherical ends induce stable

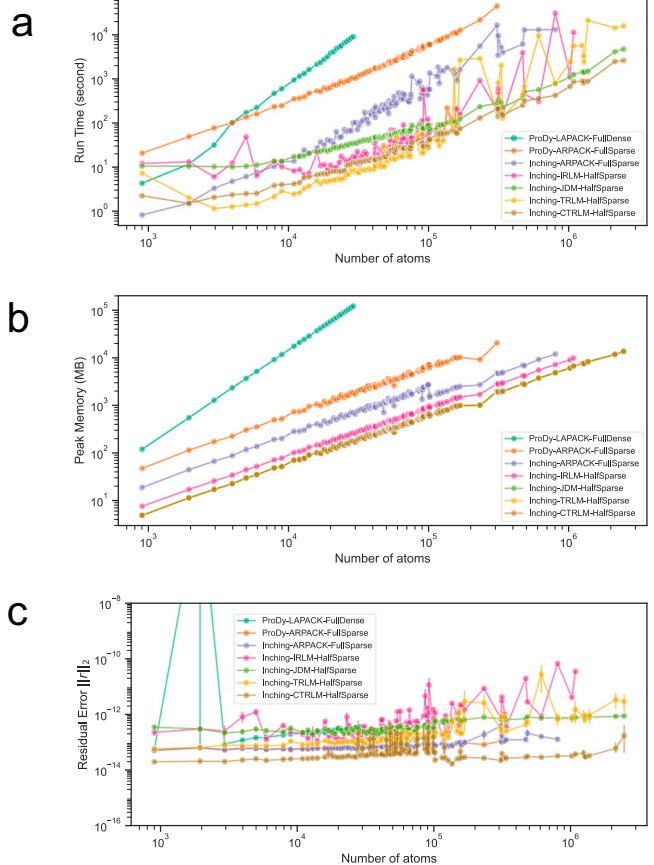

**Fig. 2 | Benchmark on throughput, memory consumption and correctness for each macromolecular structure in the benchmark dataset.** For all methods, the first 64 eigenpairs were calculated. Methods tested includes 2 ProDy methods (i.e., "ProDy-LAPACK-FullDense" and "ProDy-ARPACK-FullSparse") and 5 INCHING methods (i.e., "INCHING-TRLM-HalfSparse", "INCHING-CTRLM-HalfSparse", "INCHING-JDM-HalfSparse", "INCHING-IRLM-HalfSparse", "INCHING-ARPACK-FullSparse"). Text in the name describe the eigensolver (e.g., ARPACK, LAPACK, JDM, TRLM, CTRLM, IRLM), the access of the Hessian matrix (e.g., "Full" means that the whole Hessian matrix is accessed and stored; "Half" means that only the lower triangle of the Hessian matrix is accessed and stored.), and the storage format of the Hessian matrix (e.g., "Dense" means a 2-D array is stored in double precision; "Sparse" means a Compressed Sparse Row (CSR) format is stored in double precision). "TRLM", "CTRLM", "JDM", "IRLM" are our implementation of the Thick Restart Lanczos Method (TRLM), Chebyshev-filtered Thick Restart Lanczos Method (CTRLM), Jacobi-Davidson Method (JDM) and Implicitly Restarted Lanczos Method (IRLM). **a** Benchmark on overall run time including Hessian realization and subsequent diagonalization. The run time is wall-clock time to complete all the calculation. **b** Benchmark on peak memory consumption. Note that for "ProDy-LAPACK-FullDense", we were not able to proceed once there are more than 29 thousand atoms due to memory overflow. Note that for "ProDy-ARPACK-FullSparse", we were not able to proceed once there are more than 305 thousand atoms as it takes more than 48 h to converge. Also note that "INCHING-TRLM-HalfSparse", "INCHING-CTRLM-HalfSparse" and "INCHING-JDM-Half-Sparse" share very similar peak memory requirement. **c** Benchmark on residual error $||\mathbf{r}||_2$, defined as the 2-norm of residual vector $\mathbf{r}$ The error bar presented is the 95% confidence interval ($n = 64$) calculated from the residual error of all the eigenvalues of a macromolecular structure in the benchmark dataset. Note that for "ProDy-LAPACK-FullDense", the second macromolecular structure (PDBID: 1A8L) were not able to converge within $10^{-10}$ likely due to severe fill-ins. All programs were stopped if run time exceeds 48 h. All INCHING programs were stopped if number of restarts exceeded 15000 rounds. Source data are provided as a Source Data file.

closure of the capsid by allowing sharp bite angles at the surface[64]. With our INCHING-JDM-HalfSparse protocol, we were able to resolve its first 64 normal modes at residual error $10^{-12}$ within 1.3 h on an NVIDIA® A100 Tensor Core GPU, where the first non-zero mode corresponds to the oscillation of its hexameric surface roughly anchored at the pentamers supporting the theory of quasiequivalence[65]. In Fig. 5, the normal modes of a dilated human nuclear pore complex (NPC)[66] resolved at 50 Å were shown (EMDB: EMD14321, PDBID: 7R5J). The structure contains 4.8 million atoms and 3.5 billion non-zero entries in its Hessian. We were able to obtain the first 30 non-zero modes of this NPC structure within 12.5 h at accuracy of $10^{-12}$. In this calculation, an explicit external deflation[36] was applied to remove the first six rigid modes. We observe that while the first 14 non-zero modes mostly concern motions in the cytoplasmic and nuclear ring, the constriction of the inner ring can be observed at the fifteenth non-zero mode, reflecting the key functionally relevant conformational change. This calculation is at the edge of 80 GB of memory limit for the hardware used, though more powerful GPU systems on the market could handle even larger macromolecule superstructures. In Fig. 6, we applied the same methodology on the largest artificial DNA origami nano-structure, a DNA airplane made of 33kbps at its relaxed state[67] containing around 1.8 million atoms, including hydrogens, with a Hessian containing 1.8 billion non-zero entries. The structure has an apparent bilateral symmetry, where the joints leading to the wings are not chemically symmetric[67]. Interestingly, while its first non-zero mode presents a symmetric flop in its wings, the second non-zero mode demonstrates a complicated non-symmetric twisting motion involving its wings and stabilizers. In Supplementary Fig. 9, we also solved the first 64 modes of a 5-million pseudo-atoms coarse-grained representation of a Faustovirus capsid[68] (PDBID: 5J7V 26 million atoms resolved at 15.5 Å), illustrating the potential to incorporate coarse-graining strategies in handling systems that cannot fit into memory.

## Discussion

In recent years, there has been a significant shift in the focus of NMA methodologies, with an emphasis on accommodating larger atomic systems. The aim of an NMA is to gain insight into how some macroscopic motions involving communicating domains or subunits in macromolecular ensembles, can arise from microscopic interactions at the atomic level. In this respect, the mode shapes of NMA, which orchestrate collective motions concerning distal parts of the macromolecule, can often provide hints to understanding the biological structure at hand. Existing methods to analyze NMA can readily handle medium-sized structures with a few thousand atoms, but beyond this application of NMA is limited by computing resources and is not easily scalable by parallelization. This fundamentally restricts the molecular plasticity analysis in experimental techniques such as Cryo-ET and Cryo-EM, where macromolecular structures studied are often composed of millions of atoms and rather flexible. In these applications, fast and accurate megascale NMA would greatly facilitate modeling conformational dynamics in the refinement of the composite density maps[19–21] and the flexible fitting of atomic models[22–24] when EM volume or atomic-resolution template structures are available. Two major challenges in NMA scaling to megascale macromolecular complexes are the construction of the Hessian matrix and solution of the large-scale eigenproblem entailed. In this work, we have developed and implemented several advanced numerical recipes optimized for GPU computing, including a bandwidth-reduction algorithm and several alternative eigensolvers, to handle these challenges. This allows us to resolve normal modes, under the practical form of elastic network model, without coarse-graining the provided atomic structure for systems with several million atoms. In many cases, coarse-graining of biomolecular residues is expert-driven, and the appropriate level of granularity can be hard to determine when the chemical moiety is elongated (e.g., lipids) or is planar in shape (e.g., aromatics), especially

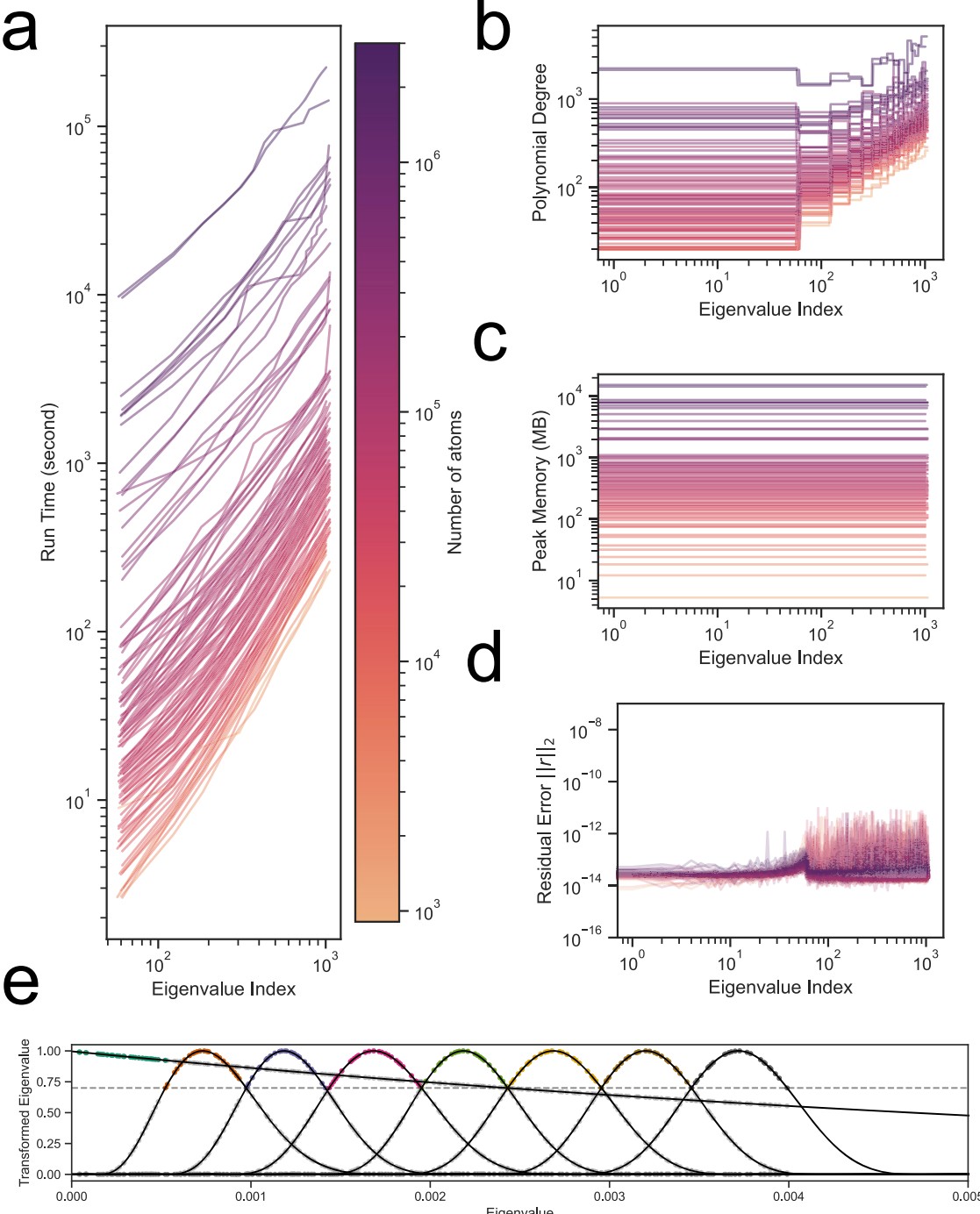

**Fig. 3 | Scaling in number of eigenmodes in Chebyshev-filtered thick restart Lanczos method.** Up to 1000 or slightly more eigenpairs were calculated using our INCHING-CTRLM-HalfSparse implementation of the Chebyshev-filtered Thick Restart Lanczos Method with radius of interaction $R_C = 8$Å. The coloring of the lines, as shown in the color bar, refers to the number of atoms in the system. **a** Run time to complete until the indexed eigenvalue (**b**) optimized polynomial degrees involved in calculating the indexed eigenvalue. **c** Peak memory consumed when calculating the spectrum slice containing the indexed eigenvalue. **d** Residual error in each of the eigenvalues. **e** Mapping of the lowest 224 eigenvalues in batches of 28 eigenvectors when low- and band-pass Chebyshev filters were applied to the HIV capsid system (PDBID: 3J3Q) without coarse-graining. Colors of the points indicate eigenvalues resolved in different batches. Source data are provided as a Source Data file.

in heterogeneous complexes comprised of protein, nucleic acid, lipid membranes, and sugars. The coarse-grained particle motions must be broadcasted into an all-atom construct if an atomic-scale understanding of the system is desired. In this respect, the implementation of an all-atom NMA, faithful to the formulation of Tirion[15], is a direct response to this shortcoming as all the atoms are now included in the

system without neglecting each of their degrees of freedom (DOF). Several recent approaches to reduce DOF of atomic systems are interesting and indispensable to further scale up though. One of the first and the most popular approaches is the Anisotropic Network Model (ANM), which consider a subset of $n_b$ atoms from the all-atom system, usually only the phosphorus P in nucleic acids and/or the

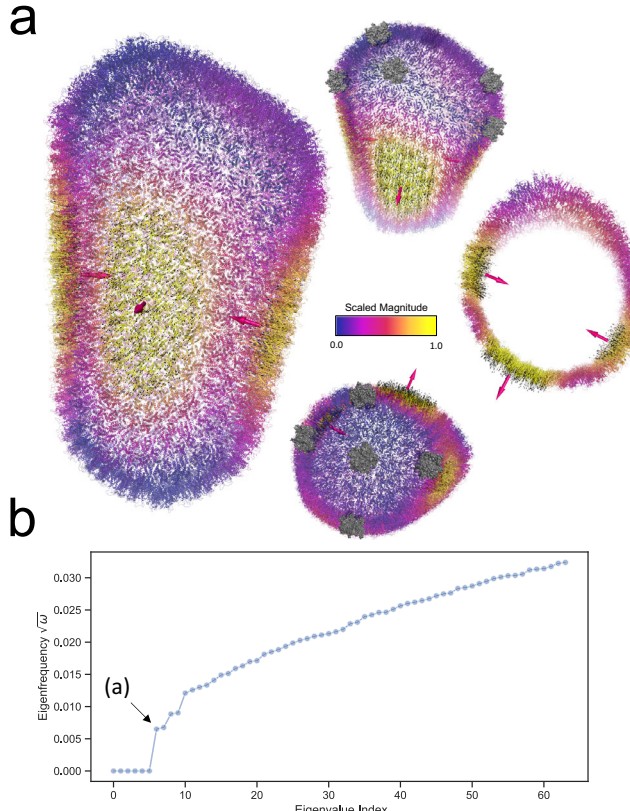

**a**

**b**

Scaled Magnitude
0.0                          1.0

**Fig. 4 | Vibrations of the mature HIV-1 capsid structure (PDBID: 3J3Q). a** The first non-zero mode of the capsid. The black arrows are the displacement field of 1000 atoms randomly chosen from those in the top 90% quantile of vibration magnitude in the eigenvector. The arrow in magenta indicates an average direction for local clusters of the displacement field. Color scale in the cartoon from blue to red indicates increasing magnitude of vibration. Note that a logistic kernel is applied to the eigenvector to control the magnitude in visualization. (See Methods). The top-right inset is a clipped view of the capsid. The bottom-right inset indicates the locations of the pentamers (Gray opaque surface). **b** The first 64 eigenfrequency of the macromolecule, the eigenfrequency is the square root of eigenvalue $\omega$. The black arrow indicates the first non-zero eigenmode displayed in (**a**). Source data are provided as a Source Data file.

backbone carbon C$\alpha$ of the proteins[15,25], thus effectively reducing $N$ for 8-10 times. Agreement of ANM with experimental data[18,25] is satisfactory, though the displacement information of the discounted atoms is lost. This issue was alleviated by the rotation-translation block (RTB) method[26], where a projection operator was developed to coarse-grain the atomic system into a system of $n_b$ rigid blocks, each with its own translational and rotational DOF, hence effectively reducing the $3N \times 3N$ Hessian matrix to a $6n_b \times 6n_b$ RTB matrix. Further refinement along this line is the Block Normal Mode (BNM)[27] method, which surrogates the initial realization of the peak-memory-consuming $3N \times 3N$ Hessian matrix; this was done by exploiting the block structure of the projection operator and by constructing only part of the sparse full atomic Hessian on-the-fly. Very recently, it was also shown that an extrapolation of RTB modes can effectively predict nonlinear motions at large amplitudes[69]. Depending on the granularity of the system, more than tenfold reduction in the size of the Hessian matrix can be achieved. However, the cost of diagonalization can still be very prohibitive.

In this respect, parallel computing on GPU, as we implemented in this work, is an effective way to amortize the cost at a fundamental level, without compromising accuracy or atomistic detail. We also expect that advances in GPU hardware[70] in memory and processing

rate, integrated with techniques in solving large-scale eigenproblems, in particular recent trends in spectrum slicing[34,52–57,71], would eventually allow even faster NMA calculations on systems exceeding 20 million atoms, such as structure of a Faustovirus[68] available in low resolution (PDBID: 5J7V, 15.5 Å). Nonetheless, given the applicability of NMA in biological systems, we believe our framework will be useful in the exploration of megascale structural dynamics. The growing complexity of the megacomplexes resolved by Cryo-EM and/or cryo-ET now calls for methods that can rapidly capture conformational and functional plasticity, potentially as an intrinsic part of the refinement pipeline. Such methods will play a crucial role in advancing our understanding of these macromolecular machines.

## Methods

### Normal mode analysis of an elastic network model

The Normal Mode Analysis (NMA) is a classic approach to derive motions from static structures at local minima of a potential energy surface. The potential energy $V$ is dependent on the conformation $\mathbf{X} \in \mathbb{R}^{3N}$ for a structure with $N$ atoms at time $t$. Without loss of generality, we begin with a particular conformation $\mathbf{X}^{(0)} \in \mathbb{R}^{3N}$. For small displacements, we may then tolerate a second-order Taylor expansion,

$$V\left(\mathbf{X}|\mathbf{X}^{(0)}\right) = V\left(\mathbf{X}^{(0)}|\mathbf{X}^{(0)}\right) + \nabla V\left(\mathbf{X}^{(0)}|\mathbf{X}^{(0)}\right)\left(\mathbf{X} - \mathbf{X}^{(0)}\right) \\ + \frac{1}{2}\left(\mathbf{X} - \mathbf{X}^{(0)}\right)^T \nabla^2 V\left(\mathbf{X}^{(0)}|\mathbf{X}^{(0)}\right)\left(\mathbf{X} - \mathbf{X}^{(0)}\right) + \dots \tag{1}$$

By choosing the energy level $V(\mathbf{X}^{(0)}|\mathbf{X}^{(0)}) = 0$ and assuming local minimum $\nabla V(\mathbf{X}^{(0)}|\mathbf{X}^{(0)})^T = 0$, we are left with the quadratic form,

$$V_{NMA} = \frac{1}{2}\Delta\mathbf{X}^\mathbf{T}\mathbf{H}\Delta\mathbf{X} \tag{2}$$

where $\mathbf{H} =: \nabla^2 V(\mathbf{X}^{(0)}|\mathbf{X}^{(0)})^T \in \mathbb{R}^{3N \times 3N}$ is the Hessian matrix and $\Delta\mathbf{X} =: (\mathbf{X} - \mathbf{X}^{(0)}) \in \mathbb{R}^{3N}$ is the deviation from the minimum. We will defer the layout of $\mathbf{H}$ to the next section when the form of potential energy $V$ is defined and proceed to discuss the outcome of an NMA. Substituting $V_{NMA}$ into the equations of motion gives a partial differential system

$$\mathbf{M}\frac{d^2\Delta\mathbf{X}}{dt^2} = -\mathbf{H}\Delta\mathbf{X} \tag{3}$$

And, the general solution $\Delta\mathbf{X} = \mathbf{Q}e^{-i\omega t}$ gives a generalized eigenproblem

$$\mathbf{HQ} = \mathbf{MQ}\Omega \tag{4}$$

Where the eigenvector matrix $\mathbf{Q} \in \mathbb{R}^{3N \times 3N}$ can be viewed as an eigentensor $\mathbf{Q} \in \mathbb{R}^{3N \times N \times 3}$. The physical meaning of this eigentensor $\mathbf{Q}$ is that each of its atom slice (the second index) gives a displacement vector, hence $3N$ linearized mode shapes, i.e., the collective motions, of the static structure $\mathbf{X}^{(0)}$ can be obtained. In NMA, the first 6 modes will always have zero eigenvalues corresponding to rigid translational and rotational displacements. The eigentensor $\mathbf{Q}$ is also the best displacement under orthonormality constraint $\min_Q V_{NMA} s.t. \mathbf{Q}^\mathbf{T}\mathbf{Q} = \mathbf{I}$. Eigenfrequency $\sqrt{\omega_i}$ is the square root of the eigenvalue $\omega_i$.

A variety of $V$ exists. In the standard NMA, potential energies from a detailed all-atom forcefield can be used. However, this approach suffers from the tedious, and sometimes virtually unachievable, requirement of energy minimization[10]. This requirement was surrogated in the work of Tirion[15], where the potential energy of an atomic system was considered as an elastic network model (ENM) connecting

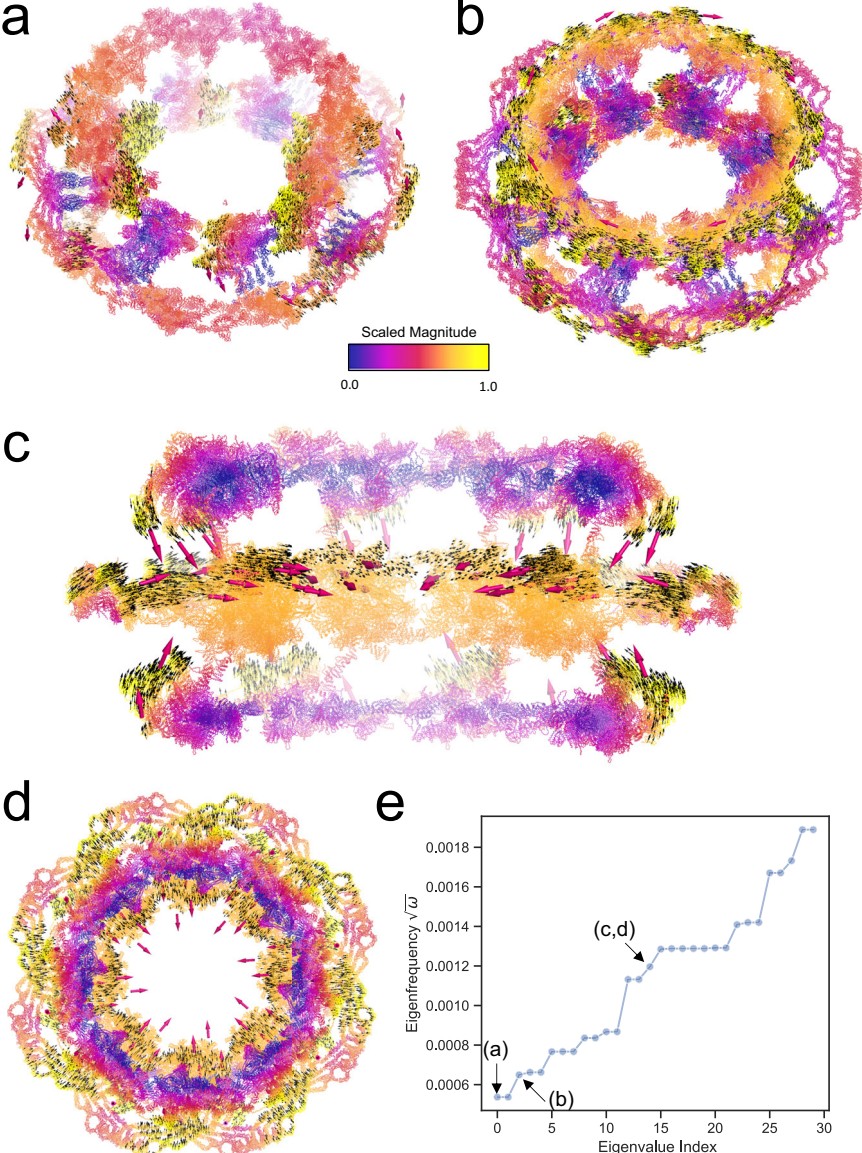

**Fig. 5 | Vibrations of the dilated human Nuclear Pore Complex (NPC).** Overview of motions in several representative non-zero modes. **a** The first non-zero normal mode (indexed as Mode 0) (**b**) the third non-zero normal mode (indexed as Mode 2) (**c**) clipped view of the fifteenth non-zero normal mode (indexed as Mode 14) (**d**) bird-eye view of the fifteenth non-zero normal mode (indexed as Mode 14) The arrows in magenta indicate an average directions for the local clusters of the displacement field, while the tiny black arrows show more detailed displacement fields of 10,000 atoms randomly chosen from those in the top 80% quantile of vibration magnitude in the eigenvector. Color scale in the cartoon from blue to red indicates increasing magnitude of vibration. Note that a logistic kernel is applied to the eigenvector to control the magnitude in visualization. (See "Methods"). **e** The first 30 non-zero eigenfrequencies of the macromolecule, the eigenfrequency is the square root of eigenvalue $\omega$. The black arrows indicate the first non-zero eigenmode displayed in (**a**), (**b**), (**c**) and (**d**). Source data are provided as a Source Data file.

atoms $i$ and $j$

$$V(\mathbf{X}) = : \sum_{ij} \frac{1}{2} k_{ij} \left( r_{ij} - r_{ij}^{(0)} \right)^2 \tag{5}$$

Where $r_{ij}$ is the variable Euclidean distance between atom pairs; $r_{ij}^{(0)}$ is the equilibrium Euclidean distance from $X^{(0)}$. Importantly, a Heaviside function $k_{ij}$, parameterized on $r_{ij}^{(0)}$, was used to eliminate the long-range interactions beyond the threshold $R_C$. This can also be considered as applying an adjacency matrix $k_{ij} \in K$ to the network.

$$k_{ij}\left( r_{ij}^{(0)} \right) = \begin{cases} 1, & r_{ij}^{(0)} \leq R_C \\ 0, & r_{ij}^{(0)} > R_C \end{cases} \tag{6}$$

As in the formulation of Tirion, a unified mass $\mathbf{M} = \mathbf{I}$ were taken, hence reducing the generalized eigenproblem to a standard eigenproblem.

$$\mathbf{HQ} = \mathbf{Q}\boldsymbol{\Omega} \tag{7}$$

**Permuting the Hessian to produce graph isomorphs**

In our later exposition, we will permute the Hessian matrix such that the resultant matrix is locally dense. Observe that a similarity transform of $\mathbf{H}$ with permutation matrix $\mathbf{P} \in \mathbb{R}^{3N \times 3N}$ preserves the eigenvalues $\boldsymbol{\Omega}$ as

$$(\mathbf{PHP^T})(\mathbf{PQ}) = (\mathbf{PQ})\boldsymbol{\Omega} \tag{8}$$

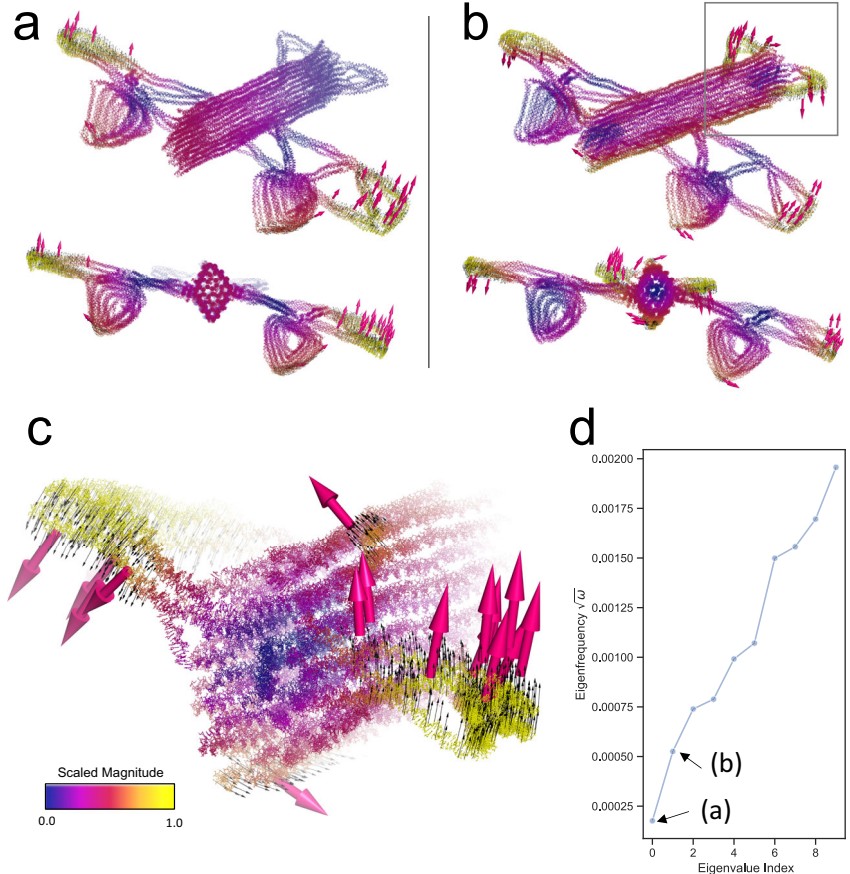

**Fig. 6 | Vibrations of a DNA origami airplane.** Overview of motions in the first two non-zero modes of the airplane. **a** The first non-zero eigenmode. **b** The second non-zero eigenmode; the square border indicates the stabilizer, which is also shown in (**c**). Color scale in the cartoon from blue to red indicates increasing magnitude of vibration. Note that a logistic kernel is applied to the eigenvector to control the magnitude in visualization. (See "Methods"). The bottom inset show the airplane along its apparent bilateral symmetry axis. **c** A zoom into the motion of the stabilizer in the second non-zero mode. The black arrows are the displacement field of 1000 atoms randomly chosen from those in the top 90% quantile of vibration magnitude in the eigenvector. The arrow in magenta indicates an average direction for local clusters of the displacement field, which also correspond to (**a**). **d** The first 10 non-zero eigenfrequencies of the macromolecule, the eigenfrequency is the square root of eigenvalue $\omega$. Note that explicit external deflation was applied to remove the rigid modes. Source data are provided as a Source Data file.

and the eigenvector $\mathbf{P^T P Q}$ of the original problem can be recovered by simply applying inverse of the unitary permutation matrix, which is its transpose. Importantly, if we restrict the permutation to atom ordering, i.e., permuting every 3 consecutive indices as an immutable group in the $3N$ indices, then the resultant Hessian is a representation of a graph isomorph of the original elastic network, where the bijection is provided by correspondence between the original atom ordering and the permuted atom ordering. This property means that the similarity transform can be computed by simply initiating with a permuted coordinate and similarly $\mathbf{Q}$ can be recovered by permuting the atom slice of $\mathbf{PQ}$ using the same bijection with linear cost.

### Constructing the Hessian using batched tensor products and broadcasts

We begin with the following un-vectorized notation, where the superscript (0) is dropped for readability.

$$\mathbf{X^{(0)}} =: \left[\mathbf{X_1^{(0)}}, \ldots, \mathbf{X_n^{(0)}}\right]; \mathbf{Vec}(\mathbf{X^{(0)}}) =: \left[x_1, y_1, z_1, \ldots x_n, y_n, z_n\right] \quad (9)$$

Applying differentiations, the off-diagonal force constant block is specified by

$$\mathbf{H_{ij}} = -\frac{k_{ij}}{\left(r_{ij}^{(0)}\right)^2}\left(\mathbf{X_i^{(0)}} - \mathbf{X_j^{(0)}}\right)\left(\mathbf{X_i^{(0)}} - \mathbf{X_j^{(0)}}\right)^T$$

$$= -\frac{k_{ij}}{\left(r_{ij}^{(0)}\right)^2}\begin{bmatrix} (x_i - x_j)^2 & (x_i - x_j)(y_i - y_j) & (x_i - x_j)(z_i - z_j) \\ (y_i - y_j)(x_i - x_j) & (y_i - y_j)^2 & (y_i - y_j)(z_i - z_j) \\ (z_i - z_j)(x_i - x_j) & (z_i - z_j)(y_i - y_j) & (z_i - z_j)^2 \end{bmatrix}$$

$$\mathbf{H_{ii}} = -\sum_j \mathbf{H_{ij}} \quad (10)$$

For any matrix $\mathbf{A}$ with $n$ rows, we can define for a row in it,

$$\beta_i =: \left| i - \min_j \left( j | a_{ij} \neq 0 \right) \right| \quad (11)$$

Then the bandwidth of the matrix is $\max_i(\beta_i)$ and the profile of the matrix is $\sum_i \beta_i$, the mean bandwidth is $\frac{\sum_i \beta_i}{n}$. Clearly, due to the cutoff $R_C$, the layout of the Hessian tensor $\mathbf{H} \in \mathbb{R}^{N \times 3 \times N \times 3}$, viewing from the atom slices, will have the same bandwidth as the adjacency matrix $\mathbf{K}$ of the macromolecule. The tensor is globally sparse in most cases. This observation also applies to standard NMA as long range cut-offs were used for van der Waals, electrostatic, and hydrogen-bonding interactions[12]. In general, it is not advisable to construct a dense Hessian matrix that contains a large number of zero entries due to the quadratic storage consumption and the resultant increase in fill-ins. To construct a sparse Hessian matrix, we find the tensorial layout by Koehl[30] a good starting notation. In which, he defined the following $N \times 3$ matrix, which can also be vectorized as $3N$ elements.

$$\mathbf{U_{ij}}\left(\mathbf{X^{(0)}}\right) = : \left(0,\ldots,0,\frac{\mathbf{X_i^{(0)}}-\mathbf{X_j^{(0)}}}{r_{ij}^{(0)}},0,\ldots,0,\frac{\mathbf{X_j^{(0)}}-\mathbf{X_i^{(0)}}}{r_{ij}^{(0)}},0,\ldots,0\right)$$ (12)

The distance $r_{ij}^{(0)}$ can be precomputed and filtered by $k_{ij}(r_{ij}^{(0)})$ for pairs within $R_C$. Then, the off-diagonal block of the entire Hessian tensor is given as a sum of Kronecker products.

$$\mathbf{H_{ij}} = \sum_i \sum_j k_{ij} \mathbf{U_{ij}} \otimes \mathbf{U_{ij}^T}$$ (13)

More importantly, when Hessian-vector multiplication $\mathbf{Hw}$ is accessed in eigensolvers, the following can be done separately for each $(i,j)$ pairs.

$$\mathbf{Hw} = \sum_{ij} k_{ij}\left(\mathbf{U_{ij}} \otimes \mathbf{U_{ij}^T}\right)\mathbf{w} = \sum_{ij} k_{ij}(\mathbf{U_{ij}w})\mathbf{U_{ij}}$$ (14)

Given explicit indexing for locations of the non-zeros $k_{ij}$, the above Hessian-vector product can be effectively parallelized on a hyperthreaded computer.

In our implementation, the tensorial representation above was refined to fully exploit on-chip parallelism on GPU and to avoid explicit index loading which tends to thread divergence on GPU. To facilitate discussion, we define a difference matrix $\mathbf{G_{ij}} \in \mathbb{R}^{N \times 3}$

$$\mathbf{G_j}\left(\mathbf{X^{(0)}}\right) = : \left(\mathbf{0},\ldots,\mathbf{0},\mathbf{X_{b_3}^{(0)}}-\mathbf{X_j^{(0)}},\ldots,\mathbf{X_i^{(0)}}-\mathbf{X_j^{(0)}},\ldots,\mathbf{X_{b_4}^{(0)}}-\mathbf{X_j^{(0)}},\mathbf{0},\ldots,\mathbf{0}\right)$$ (15)

Supposed, we know $|b_3 - b_4|$ is the bandwidth of an off-diagonal batch of $H$, then, for two contiguous atom indices $(b_1,b_2]$ and $(b_3,b_4]$, the batch $\mathbf{G} \in \mathbb{R}^{3|b_1-b_2| \times 3|b_3-b_4|}$ can be given by the four index corners $(b_1,b_2,b_3,b_4)$.

$$\mathbf{G_{(b_1,b_2,b_3,b_4)}} = \sum_{i\in(b_1,b_2]}\sum_{j\in(b_3,b_4]} \mathbf{G_i} \otimes \mathbf{G_j^T}$$ (16)

In practice, $(b_3,b_4]$ could include intervals that lack neighboring atoms for the range $(b_1,b_2]$, but the indexing slice $(b_3,b_4]$ can be further refined into multiple contiguous indexing slices to eliminate unnecessary calculations. Nevertheless, filtering for non-zero elements in the batch can be done by observing Eq. (10) in that the squared interatomic distance is given by the trace of each block $(r_{ij}^{(0)})^2 = tr(\mathbf{G_{ij}})$, then we can consider the following block-wise operations, readily

parallelizable on GPU by in-place tensorial broadcasts.

$$Y_{ij} = tr(\mathbf{G_{ij}})$$

$$K_{ij} = \begin{cases} 1, & Y_{ij} \leq R_C^2 \\ 0, & Y_{ij} > R_C^2 \end{cases}$$

$$Y_{ij} = \begin{cases} Y_{ij}^{-1}, & \forall i \neq j \\ 0, & \forall i = j \end{cases}$$ (17)

$$\mathbf{H_{(b_1,b_2,b_3,b_4)}} = \left(\left(\mathbf{K_{(b_1,b_2,b_3,b_4)}} \odot \mathbf{Y_{(b_1,b_2,b_3,b_4)}}\right) \otimes \mathbf{1_{3\times 3}}\right) \odot \mathbf{G}$$

The $\odot$ is the Hadamard product operator; $\mathbf{1_{3\times 3}}$ is the all-ones matrix presenting the broadcast. This is followed by row-sum in the diagonal $\mathbf{H_{ii}} = -\sum_{\mathbf{j}\in(\mathbf{b_3},\mathbf{b_4}]}\mathbf{H_{ij}}$ accordingly. Only the lower triangle is incrementally stored in the Column-Sparse-Row (CSR) format for subsequent calculation, hence consuming $O(9p)$ memory for the data content; $p$ is the number of non-zero pairwise interaction for $i \geq j$. The dense batch $\mathbf{G_{(b_1,b_2,b_3,b_4)}}$ presented above will require $O(9|b_1 - b_2||b_3 - b_4|)$ erasable memory, but we will show later that the local bandwidth $|b_3 - b_4|$ can be reduced.

## Bandwidth and layout of the Hessian

Obviously, the cost of the parallel computation will depend on the local bandwidth $|b_3 - b_4|$ of the batch $(b_1,b_2]$. The Hessian obtained with the default ordering of atoms can result in very large bandwidths with non-uniform patterns meaning interactions among sequentially distal parts within the tertiary structure or individual chains within a quaternary structure. A hypothetical minimal example is a macromolecule with 3 peptide chains α, β and γ, where only α-γ and β-γ interactions were found, but not α-β; in this case, the order α-β-γ will have a much larger bandwidth than α-γ-β. Many reasons can attribute to this observation. An example found in the benchmark set is a chimeric Sesbania mosaic virus coat protein[72](PDBID: 4Y5Z) composed of 12 sets of pentamers arranged on the vertices of an icosahedron, where each protein chain in the pentamer interacts with 7 other chains within 8 Å. (See Supplementary Fig. 1) A systematic order to build up this icosahedron, as done by the authors, is to place a pentamer on each of the four corners of the three orthogonal golden rectangles, done one rectangle after another starting at the shorter sides of the golden rectangles. (See Supplementary Fig. 1a) In this case, pentamers on the shorter sides interact within 8 Å among each other, but pentamers on the long sides of the same rectangle do not. This creates exactly the situation where the aforementioned α-β-γ order arises. Indeed, the vertices on a icosahedron can never be clustered satisfactorily and there are specialized algorithm to calculate their vibrational dyanmics[73]. Besides, due to the inconsistent presence of water molecules, the exact symmetry is destroyed in the crystal structure, which is commonly encountered. Nevertheless, the degeneracies in modes due to the apparent symmetry of this macromolecule is captured in our program. (See Supplementary Fig. 1d) There are also cases where no meaningful sequential ordering is possible when intact chemical structures (e.g. cofactors, ions, water, lipids) are intercalated between sequentially distal parts of the polymer. A practical example in the benchmark set is a PsbM-deletion mutant of photosystem II[74] (PDBID: 5H2F), where elongated lipids and cofactors are integral part of the macromolecule complex surrounded by several protein chains. In this case, it is not obvious as for how to rearrange the atoms to reduce the bandwidth, but we can show that the Hessian can always be permuted to reduce bandwidth by the algorithm described in the next section. (See Supplementary Fig. 2)

## Bandwidth reduction of the Hessian matrix

Finding a permutation to minimize the bandwidth is NP-complete[41], but a reduction of bandwidth and its overall profile can be approximated by algorithms such as the Reverse Cuthill McKee (RCM)

algorithm. The RCM is a greedy approximation with a breadth-first level structure. The input of a standard RCM algorithm is an adjacency graph, and its outputs is a permuted node ordering. In RCM, starting from a peripheral node with the lowest degree of connection, adjacent unvisited nodes are collected as a level structure and re-ordered by their degree of connection. Provided that an adjacency matrix $\mathbf{K}$ is precomputed, the time complexity of the RCM is bounded by $O(4|E| + 2cm|E| + N)$, where $m = : \max \sum_j K_{ij}$ is the maximum degree among all nodes and $2|E|$ is the number of edges in $\mathbf{K}$. A detail proof is in reference[43]. The term $4|E|$ is referring to $2|E|$ operations to determine the degree of each node plus another $2|E|$ operations to sweep through the adjacency matrix to locate the neighbors. The term $2cm|E|$ refers to the insertion sorting of the degree in retrieved neighbors. The last term $N$ is due to reversal of order. The input of RCM is an adjacency matrix $\mathbf{K}$. For 3-D coordinates, this may be obtained by a cell-linked list data structure in $O(27Nn_c)$ where $n_c$ is the average number of particles per cell in volume $R_c^3$ or simply an all-to-all calculation as done in ProDy2.4 for small-sized systems. Therefore, a standard RCM algorithm operating on 3-D coordinate data will require a total time complexity of $O(27Nn_c + 4|E| + 2cm|E| + N)$ and ideally a $O(mN)$ memory to operate due to the adjacency matrix. To surrogate the storage of the adjacency matrix, we supplemented the RCM with a k-D tree data structure[47], which takes the coordinate data $\mathbf{X}^{(0)}$ directly as input. The algorithm is labeled as 3DRCM to avoid confusion with the standard RCM. A pseudocode of our 3DRCM implementation is provided in the Supplementary Information as Algorithm 1. A 3-D tree is a balanced space-partitioning binary tree, which can be constructed and stored by finding and storing hyperplanes that split the number of points in halves. As such, the construction of a 3-D tree takes $O(N \log N)$ time and $O(3N)$ space. On a 3-D tree, each range search for adjacent neighbors within threshold distance takes $O(3N^{2/3})$[75]. At this stage, the total time complexity of 3DRCM appears to be $O(2*3N^{\frac{2}{3}} + 2cm|E| + N)$, where the first term $2*3N^{\frac{2}{3}}$ is due to the collection of degree in each node and the search for neighbor, but we will show that in both 3DRCM and RCM the term $O(2cm|E|)$ is dominating. A tighter bound on $O(2cm|E|)$ in the context of NMA can be obtained as follows. For a stable macromolecule without atomic clashes, the shortest interatomic distance is due to covalent bonds at around 1Å. The Kepler's conjecture proven by Hales[76] states that the maximum volume ratio occupied by equidistant sphere packing is $\frac{\pi}{3\sqrt{2}} < 0.7405$. Hence, the maximum number of atoms allowable without clashes within radius $R_C$ is $\rho = : R_C^3 \frac{\pi}{3\sqrt{2}} \geq m$. This packing density $\rho$ presents an upper bound to the number $m$ of non-zero entries in a row on the adjacency matrix. This bounds the total number of edges as

$$\rho N \geq mN > 2|E| \tag{18}$$

For an 8 Å radius, there can only be less than 380 atoms in absence of clash. Therefore, the time complexity of 3DRCM and RCM in the context of NMA are bounded by $O(6N^{\frac{2}{3}} + c\rho^2 N + N)$ and $O(27N\rho + 2\rho N + c\rho^2 N + N)$ respectively. Taking $c = 1$, the 3DRCM is dominated by the pseudolinear term $O(\rho^2 N)$ unless $N$ is beyond 3.7 million atoms. From experience, for a system containing 2.4 million atoms, computing the permutation ordering with 3DRCM and the calculation of the Hessian took only 5.2 min and 6.9 min in wall-clock time respectively. See Supplementary Fig. 4. Hence, for most practical purpose, the trade-offs in time complexity in 3DRCM compared to RCM is negligible.

## Low-complexity Krylov-subspace eigensolvers

In this work we have implemented several iterative methods[31,32,49,58] to solve large eigenvalue problems on the GPU. Specifically, in NMA, we are mostly interested in the smallest non-zero eigenpairs. The Hessian matrix of concern is a sparse positive semidefinite real symmetric matrix without weak diagonal dominance. To facilitate communication, we adopt more generic notations here. The matrix of concern is notated as $\mathbf{A} \in \mathbb{R}^{n \times n}$; the exact and the approximate eigenpairs are $(\boldsymbol{\Lambda} \in \mathbb{R}^{m \times m}, \mathbf{U} \in \mathbb{R}^{n \times m})$ and $(\widetilde{\boldsymbol{\Lambda}} \in \mathbb{R}^{m \times m}, \widetilde{\mathbf{U}} \in \mathbb{R}^{n \times m})$ respectively; $n$ and $m$ denotes the dimension of the basis set and the number of basis respectively. The standard eigenproblem is thus $\mathbf{AU} = \mathbf{U}\boldsymbol{\Lambda}$ and we assumed orthonormality among the exact eigenvectors. The approximate $\widetilde{\lambda} \in \widetilde{\boldsymbol{\Lambda}}$ can be obtained from the Rayleigh quotient $\widetilde{\lambda} = \widetilde{\mathbf{u}}^{\mathbf{T}} \mathbf{A} \widetilde{\mathbf{u}}$ and the residual $\mathbf{r} = : \mathbf{A}\widetilde{\mathbf{u}} - \widetilde{\lambda}\widetilde{\mathbf{u}}$ can always be evaluated by its 2-norm, the residual error $||\mathbf{r}||_2$.

In the following paragraphs, we will progressively introduce two branches of iterative methods implemented, namely the Hermitian Lanczos Methods followed by the Jacobi-Davidson Method. Only rationales and key equations were presented. A common central idea is to find $\widetilde{\mathbf{u}}$ by refining an initial guess $\mathbf{v_0} \in \mathbb{R}^n$ to build up an orthonormal basis $\mathbf{V} \in \mathbb{R}^{n \times m}$ in the Krylov subspace

$$\mathcal{K}_k(\mathbf{A}, \mathbf{v_0}) = : span\{\mathbf{v_0}, \mathbf{A^1v_0}, \mathbf{A^2v_0}, \ldots, \mathbf{A^{k-1}v_0}\} \tag{19}$$

The residual is minimized when the Galerkin condition

$$\left(\mathbf{Ve_j}\right)^{\mathbf{T}} \left(\mathbf{A}\widetilde{\mathbf{u}} - \widetilde{\lambda}\widetilde{\mathbf{u}}\right) = 0 \forall j \in (0, \ldots, m-1) \tag{20}$$

is satisfied, where $\widetilde{\mathbf{u}} = : \mathbf{Vy}$. The $\mathbf{y} \in \mathbb{R}^{m \times 1}$ is an unknown component to combine $\mathbf{V}$, but if $\mathbf{V} \in \mathscr{K}_k$ has its orthonormality maintained satisfactorily, for example by incorporating the Modified Gram Schmidt algorithm (MGS), then $\mathbf{y}$ is the eigenvector of a symmetric tridiagonal eigenproblem[77] of a much smaller size $m \times m$

$$\left(\mathbf{V^TAV}\right)\mathbf{y} = \widetilde{\lambda}\mathbf{y}, \tag{21}$$

and $\widetilde{\mathbf{u}} \in \mathbb{R}^n$ can be recovered by definition. This orthogonal projection technique, an example of Rayleigh-Ritz process, is common to all three methods implemented.

**Technical remarks.** Given the recurring application of certain techniques in the implemented methods, an assortment of technical remarks is presented here before progressing further. The MGS can be applied more than once to prevent loss of orthogonality due to float point roundoffs and the procedure is called a full reorthogonalization (FRO). A pseudocode of orthogonalizing a vector against a basis with MGS, and similarly with an Iterative Classical Gram-Schmidt (ICGS) algorithm is provided in the Supplementary Information as Algorithm 2 and Algorithm 3. The low-complexity $O(n^3)$ cost of the matrix-vector multiplications in producing $\mathscr{K}_k$ can be amortized by parallelisms in GPU. When only the lower triangle $\mathbf{L}$ of $\mathbf{A}$ is available, the multiplication with an arbitrary vector $x$ can be done as a sum decomposition $\mathbf{Lx} + \mathbf{L^Tx} - \mathbf{Dx}$, where $\mathbf{D} = : diag(\mathbf{A})$. In our implementation, a custom kernel that utilizes the SpMV algorithm in CuSPARSE is written to accommodate the availability of the lower triangle, halving the memory requirement to access the Hessian. In all cases, the convergence rate of the $i$-th eigenpair is dictated by the ratio of adjacent exact eigenvalues $\left|\frac{\lambda_i}{\widetilde{\lambda}_{i+1}}\right| \leq 1$, the smaller the better. Note that $\mathbf{A}$ was shifted upward as $\mathbf{A} + \mathbf{I}$ to avoid poor condition number. Several strategies to improve the convergence rate for the smallest eigenpairs includes (1) the Shift-and-Inverse technique, where the Ritz pairs of $(\mathbf{A} - \sigma\mathbf{I})^{-1}$ closest to $\sigma$ was sought instead and the inverse is incorporated into the matrix-vector product by solving for $\mathbf{v}^{(k+1)}$ in $(\mathbf{A} - \sigma\mathbf{I})\mathbf{v}^{(k+1)} = \mathbf{v}^{(k)}$, (2) the implicitly shifted QR algorithm by Francis[78], where multiple shifts were applied to the subspace iteration problem typically concerning a small eigenproblem (See part b of Algorithm 5) and (3) filter diagonalization[33,34,57] that magnifies convergence rate for regions of a

spectrum (See Algorithm 4,8 and 9 and later sections). Finally, the six exact eigenvectors $\mathbf{U}_6$ corresponding to the rigid modes can be removed from $\mathscr{K}_k$ by an explicit external deflation[36]; the dense deflated matrix $\mathbf{A} - \sigma_H \mathbf{U}_6 \mathbf{I}_6 \mathbf{U}_6^{\mathbf{T}}$, with eigenvalues corresponding to $\mathbf{U}_6$ raised to $\sigma_H$, is not stored but incorporated into the matrix-vector multiplications; the sparse $\mathbf{U}_6$ is stored in CSR format. This is implemented in all three methods and can be optionally called.

**Hermitian Lanczos method (HLM).** The $\mathbf{V} \in \mathscr{K}_k$ described earlier can be obtained iteratively as $\mathbf{v}_{k+1}$ by projecting away components of previously obtained $\mathbf{v}_i \forall i \in 0 \ldots k$ from the current matrix-vector product $\mathbf{A} \mathbf{v}_k$. In the passing, the smaller eigenproblem $\mathbf{T} := \mathbf{V}^{\mathbf{T}} \mathbf{A} \mathbf{V}$ is also produced. Importantly, for a symmetric matrix $\mathbf{A}$ and $\mathbf{V} \in \mathscr{K}_k$, the $\mathbf{T}$ is symmetric tridiagonal. Therefore, the recursion to obtain $\mathbf{v}_{k+1}$ simplifies to three terms

$$\mathbf{v}_{k+1} = \mathbf{A} \mathbf{v}_k - \beta_k \mathbf{v}_{k-1} - \alpha_k \mathbf{v}_k \tag{22}$$

Where $\alpha_k := \mathbf{v}_k^{\mathbf{T}} \mathbf{A} \mathbf{v}_k$ and $\beta_k := \mathbf{v}_{k-1}^{\mathbf{T}} \mathbf{A} \mathbf{v}_k$ are the main diagonal and the first subdiagonal of $\mathbf{T}$. This is the essence of the HLM proposed by Lanczos[79]. Collecting the recursion and a rearrangement reveals the Lanczos factorization stopping at the $k$-th step.

$$\mathbf{A} \mathbf{V}_k = \mathbf{V}_k \mathbf{T}_k + \beta_k \mathbf{v}_{k+1} \mathbf{e}_k^{\mathbf{T}} \tag{23}$$

A pseudocode of a p-step Lanczos Factorization is provided in the Supplementary Information as Algorithm 4. Note that the approximate eigenvectors obtained from the HLM will converge to those with extremal eigenvalues, i.e., both the largest and the smallest eigenpairs, but only the smallest eigenpairs are wanted in our problem setting. Further, while the basis $\mathbf{V}_k$ builds up in each iterative step, eventually we will experience overflow in storage, likely before convergence. To address these concerns, two practical variants of the HLM with different restarting techniques were proposed.

- **Implicitly Restarted Lanczos Method (IRLM).** The IRLM algorithm was proposed by Lehoucq and Sorensen[49]. In IRLM, the Lanczos factorization is supplemented with an implicitly shifted QR algorithm by Francis[78]. The purpose is to shift the smaller eigensystem $\mathbf{T}^{(j-1)}$ at each restart round $j$ by the current approximation of the $m - k$ unwanted eigenvalues such that the convergence rate of the $k$ wanted smallest eigenvalues is improved. The only information we need to determine these shifts is the sorted approximated eigenvalues and the number of wanted eigenvalues we desired. As a result, after obtaining each QR factorization $\mathbf{Q}^{(j)} \mathbf{R}^{(j)} = \mathbf{T}^{(j-1)} - \tilde{\lambda}_j \mathbf{I}$, the Lanczos factorization in Eq. (23) is modified as

$$\mathbf{A} \widehat{\mathbf{V}}_k = \widehat{\mathbf{V}}_k \widehat{\mathbf{T}}_k + \mathbf{v}_{k+1} \widehat{\mathbf{b}} \tag{24}$$

where $\widehat{\mathbf{b}} = \mathbf{Q}^{(j)\mathbf{T}} \mathbf{e}_k$ and $\widehat{\mathbf{T}} = \mathbf{R}^{(j)} \mathbf{Q}^{(j)} + \tilde{\lambda}_j \mathbf{I}$ and the first $k$ orthonormal basis is updated as $\widehat{\mathbf{V}}_k = \mathbf{V}_k \mathbf{Q}^{(j)}$. This is equivalent to performing $m - k$ simple polynomial filterings[80]. In our implementation, to promote parallelism, the $m - k$ QR factorizations was performed before a cumulative update of the matrices; this is followed by a full reorthogonalization of the updated basis $\mathbf{V}_k$. Note that in IRLM the residual error $||\mathbf{r}||_2$ is bounded by the $\beta_{k+1}$, a result due to Paige[81] such that $||\mathbf{r}||_2$ needs not be computed. A pseudocode of our IRLM implementation is provided in the Supplementary Information as Algorithm 5.

- **Thick-Restart Lanczos Method (TRLM).** The TRLM algorithm was proposed by Wu and Simon[58]. In TRLM, the smaller eigenproblem $\mathbf{T}$ is again solved with its eigenpairs sorted, but rather than performing implicit shift, the $\mathbf{T}$ is projected onto its wanted eigenvectors $\mathbf{Y}_k$ as $\widehat{\mathbf{T}} = \mathbf{Y}_k^{\mathbf{T}} \mathbf{T} \mathbf{Y}_k$ and the basis updated to $\widehat{\mathbf{V}} = \mathbf{V} \mathbf{Y}_k$

accordingly. As a result, the Lanczos factorization (in Eq. (23)) is modified as

$$\mathbf{A} \widehat{\mathbf{V}}_k = \widehat{\mathbf{V}}_k \widehat{\mathbf{T}}_k + \beta_k \mathbf{v}_{k+1} \mathbf{e}_k^{\mathbf{T}} \mathbf{Y} \tag{25}$$

When we want to find the next $\mathbf{v}_{k+1}$ by orthogonalizing $\mathbf{A} \mathbf{v}_k$ against the previous $\widehat{\mathbf{V}}_k$, these transforms allow us to compute it with $\beta_{k-1}(\mathbf{Y}^{\mathbf{T}} \mathbf{e}_{k-1})$ known from the restart

$$\widehat{\mathbf{v}}_{k+1} = \mathbf{A} \widehat{\mathbf{v}}_k - \widehat{\alpha} \widehat{\mathbf{v}}_k - \widehat{\mathbf{V}}_{k-1} \beta_{k-1}(\mathbf{Y}^{\mathbf{T}} \mathbf{e}_{k-1}) \tag{26}$$

Full reorthogonalization was done in the basis $\widehat{\mathbf{V}}_k$. A pseudocode of our TRLM implementation is provided in the Supplementary Information as Algorithm 6.

**Jacobi-Davidson method (JDM).** The JDM algorithm was proposed by Fokkema, Sleijpen and Van der Vorst[32]. Similar to the HLM, the JDM also considers approximations in the Krylov subspace, but rather than refining the approximations with a Lanczos factorization, it attempts to solve the following equation

$$\mathbf{A}(\widetilde{\mathbf{u}} + \mathbf{z}) = (\widetilde{\lambda} + \eta)(\widetilde{\mathbf{u}} + \mathbf{z}) \tag{27}$$

where an approximate solution pair $(\eta, \mathbf{z})$ to correct the approximate eigenpair $(\widetilde{\lambda}, \widetilde{\mathbf{u}})$ can be obtained by incorporating the Galerkin condition $(\widetilde{\mathbf{u}} + \mathbf{z})^{\mathbf{T}} \mathbf{z} = 0$ into projectors resulting in the following correction equation

$$(\mathbf{I} - \widetilde{\mathbf{u}} \widetilde{\mathbf{u}}^{\mathbf{T}})(\mathbf{A} - \widetilde{\lambda} \mathbf{I})(\mathbf{I} - \widetilde{\mathbf{u}} \widetilde{\mathbf{u}}^{\mathbf{T}}) \mathbf{z} = -\mathbf{r} \tag{28}$$

The correction equation does not need to be solved exactly such that $\mathbf{A}$ in this equation can be replaced by a preconditioner of $\mathbf{A}$ or by simply finding a solution of $\mathbf{z}$ to certain extent of precision. For the NMA eigenproblems, we cannot find a satisfactory preconditioner that is not dense such that GPU memory is not overflowed. Therefore, in our implementation, the latter strategy was adopted by restricting the number of steps or precision in the solution in the generalized minimal residual iteration (GMRES), which is also a Shift-and-Inverse example. An implementation of the GMRES taking a sparse matrix input was modified from the CuPy package. A pseudocode of our JDM implementation is provided in the Supplementary Information as Algorithm 7.

### Filter diagonalization

The Lanczos and Davidson methods can be adjusted by filters to isolate certain eigenvalues. The motivation of this Filter Diagonalization (FD) approach is to transform the spectrum such that eigenvalues from wanted intervals become dominant[33,82]. (See Fig. 6e for an illustration.) For example, to obtain the lower spectrum, the Chebyshev-Davidson Method[35] (CDM) removed the Jacobi correction in JDM (which approximates a moving rational filter) and applied a moving low-pass, fixed-degree, Chebyshev polynomial of the first kind to the input matrix. (Algorithm 8) FD can also be applied to solve for the interior rather than the extremal eigenpairs[34,57]. This can be useful when we are extracting a large amount of eigenpairs from the lower end of the spectrum, as then the interior slices of the spectrum can be obtained without storing and orthogonalizing against the filtered subspace from the very lowest end. To achieve this, a band-pass filter, which is an $M$-degree Chebyshev expansion of a Dirac-delta-like function damped by a Jackson kernel[33,83], was developed in the EVSL library[34,57] and incorporated into our INCHING-CTRLM protocol (Algorithm 6, shared with INCHING-TRLM protocol). The band-pass filter $p(t)$ were obtained following the three-term recurrence of $T_j$, the $j$-th degree Chebyshev

polynomial of the first kind.

$$T_{j+2}(t) = 2tT_{j+1}(t) - T_j(t); T_o(t) = 1; T_1(t) = t;$$
$$p(t) = \sum_{j=0}^{j=M} g_j^{(M)} \mu_j T_j(t) / \sum_{j=0}^{j=M} g_j^{(M)} \mu_j T_j(\cos(\theta)) \tag{29}$$

The coefficients $\mu_j$ in the expansion and the damping kernel $g_j^{(M)}$ were precalculated with the following equations, where $\delta_{j0}$ is the Kronecker delta and $\tilde{\pi} =: \pi/(M+2)$ and $\theta$ is the adjusted center of the arccosine of the corresponding transformed wanted interval $[\alpha_s,\beta_s]$ within $[-1,1]$. (See reference[34,57] for further details.)

$$g_j^{(M)} =: \frac{\sin((j+1)\tilde{\pi})}{(M+2)\sin(\tilde{\pi})} + \left(1 - \frac{j+1}{M+2}\right)\cos(j\tilde{\pi})$$
$$\mu_j = -\frac{1}{2}\delta_{j0} + \cos(j\theta) \tag{30}$$

The filter was applied during matrix-vector multiplication (Algorithm 9), where the spectrum of $A$ were scaled and shifted to the range of cosine $[-1,1]$ using spectral bounds from the Lanczos process (i.e. Algorithm 4) following reference[35].

## Implementation notes

In many parts of the algorithms presented above, matrix-vector multiplication, whether dense or sparse, is often the calculation bottleneck. In our implementations, this shortcoming is amortized by capitalizing on the high degree of parallelism in GPU and existing techniques built around NVIDIA® GPUs, including CUDA[61] (v. 11.6.2), cuBLAS[62], and cuSPARSE[63]. To facilitate installation, code readability and version control, the CuPy package (v. 11.5) was used as an application programming interface to access these technologies. In all the algorithms implemented, only the lower triangle of the Hessian matrix was required as input in CSR format. Our INCHING-JDM and INCHING-IRLM implementations were written with reference to the thesis of Geus[84] and the ARPACK package[50] respectively. Our INCHING-TRLM implementation is modified from the CuPy library's default, where memory footprint was optimized by exploiting sum decomposition of the lower triangle; further speedup was achieved by reducing the number of transposes done to the basis set. Where a smaller projected eigenproblem has to be solved, the calculation was done on CPU by calling 'numpy.linalg.eigh' from the NumPy (v. 1.23.5) package[85]. For methods with Chebyshev filter, the implementations in the EVSL library[57] and the Chebyshev-Davidson method[35] were referenced. When a band-pass Chebyshev filter is invoked, the converged eigenvectors are sorted again before off-loading for storage. For benchmarks with ProDy, the ProDy (v.2.4) package[48] was installed with the NumPy (v. 1.23.5) package[85] and the SciPy (v.1.8.0) package[86]. Dense eigenproblems in ProDy are solved using a LAPACK backend called through NumPy. Sparse eigenproblems in ProDy are solved using a ARPACK backend called through SciPy 'scipy.sparse.linalg.eigsh'. Benchmarks on correctness, memory and speed were conducted on a single piece of A100 NVIDIA® GPU with tensor core activated and 80GB GPU memory capacity and the AMD EPYC™ 7513 processor with 64 threads. For systems with more than 2.1 billion non-zero entries, 64-bit integers were used for indexing in the CSR format, otherwise 32-bit integers were used by default. Methods were also tested on a RTX4090 NVIDIA® GPU with 24GB memory and 64-bit indexing used throughout.

## Hyperparameters in calculations

In all cases, the following hyperparameters were used unless otherwise stated. We followed the work of Koehl[30] taking the neighborhood cutoff threshold $R_C = 8$Å for atomic systems. After performing 3DRCM, the indexing slice for each batch of atom was analyzed for presence of gaps, i.e., regions where no neighbor of the batch is present. The indexing slice was then split into multiple contiguous indexing slices by removing gaps that extend for more than 100 atoms. For INCHING-IRLM, the tolerance of error bound is set to $10^{-10}$. For both INCHING-TRLM and INCHING-JDM, the tolerance of residual error is $10^{-12}$. The correction equation in INCHING-JDM was solved using the GMRES algorithm, where only 20 steps of minimization at max were allowed. The default maximum allowed number of restarts for INCHING-IRLM, INCHING-TRLM and INCHING-JDM is 15000 steps, but for $R_C = 6$Å benchmarks, at max 30000 steps were allowed to accommodate difficult convergence in some cases. The maximum allowed basis in the Krylov subspace is three times the desired number of output eigenpairs, i.e. which is $64 \times 3 = 192$ vectors in the benchmark. The 1000 modes presented in Fig. 6 (200 modes in Supplementary Fig. 10) were calculated using INCHING-CTRLM in batches of 64 modes with 128 basis (28 modes with 56 basis) for all structures, though larger basis were affordable; external explicit deflation of the free modes were applied when low-pass filter is used. To scan through the lower end of the spectrum using INCHING-CTRLM, we begin by applying the low-pass filter to obtain the largest of the smallest $m$ eigenvalues $\alpha_s$ and set $\alpha_s - 10^{-10}$ as the left bound of the next interior wanted interval $[\alpha_s,\beta_s]$; the process continues with the band-pass filter until a desired number of eigenpairs are obtained. Note that a binary search was performed to locate $\beta_s$ such that $[\alpha_s,\beta_s]$ contains $<m$ eigenvalues for the band-pass case and $<5m$ eigenvalues for the low-pass case. The polynomial degrees were also maximized to maintain the smallest transformed eigenvalues in $[\alpha_s,\beta_s]$ as 0.7 for both filters such that convergence rate remained constant. (See Fig. 6e.). All programs were stopped if convergence was not achieved within 48 h, wall-clock time. All INCHING programs were stopped if the maximum number of restarts were exceeded. For all benchmarks with $R_C = 14$Å, 64-bit integers indexing were used for consistency.

## Benchmark coordinates

While ideally all the structures on the Protein Data Bank can be considered, we randomly selected structures at an increasing interval. For systems with less than 100 thousand atoms, structures were downloaded in the PDB format; the interval of increase is around 1 thousand atoms. For systems with more than 100 thousand atoms, structures were downloaded as mmCIF format; the interval of increase is around 10 thousand atoms. Note that while the iterative methods all work very well, some structures can contain components not connected within 8 Å threshold, resulting in more than 6 rigid modes. This happens in a small portion of PDB structures when crystallographic water(s) or protein chain(s) with no neighbor within 8 Å were scrupulously put into the crystallographic map. These structures were removed from consideration to avoid confusion. Overall, 85 structures with less than 100 thousand atoms and 31 structures with more than 100 thousand atoms were tested. This benchmark set contains 116 structures in total with number of atoms ranging from around 1 thousand atoms to 2.4 million atoms.

## Megascale atomic structures

We applied our method to some of the largest atomic objects available. The mature HIV-1 capsid structure (PDBID:3J3Q) containing around 2.4 million atoms was obtained from the Protein Data Bank[37] without any modification. The human Nuclear Pore Complex structure (EMDB: EMD14321, PDBID: 7R5J) was downloaded as a bioassembly from the Protein Data Bank[37] and chains LA,MA,NA,OA with clashing linkers were removed; the final structure contains 4.8 million atoms. The atomic structure of the 26 million atoms Faustovirus capsid (PDBID:5J7V) was downloaded from the Protein Data Bank; Cα were extracted from the structure resulting in a 5 million pseudo-atom coarse-grained representation; NMA with $R_C = 14$Å was performed with INCHING-CTRLM-HalfSparse with external explicit deflation of the 6

free modes; the first 64 non-zero modes were obtained. The atomic structure of the DNA airplane[67] was obtained from Nanobase.org[87] and the atomic coordinates were reconstructed using TacoxDNA[88]; the final structure includes hydrogens and contains 1.8 million atoms. The NMA of HIV-1 capsid structure were calculated using INCHING-JDM-HalfSparse and the same hyperparameters of numerical methods outlined above. For the DNA airplane and the NPC complex, the NMA was calculated using INCHING-JDM-HalfSparse with external explicit deflation of the 6 free modes, the number of modes to resolve, the maximum allowed basis in the Krylov subspace, the maximum number of allowed steps in GMRES and the maximum number of restarts allowed were revised to the first 30 non-zero normal modes, 120 vectors, 150 GMRES steps and 550000 restarts respectively, otherwise all other hyperparameters are the same. The megascale structures and two movies corresponding to a mature HIV-1 capsid structure (PDBID:3J3Q, Supplementary Movie 1) and a ribosome bound to elongation factor G (PDBID:4V9H, Supplementary Movie 2) were displayed using PyMOL (v2.4.1).

**A logistic kernel to visualize atomic motion**

In all the iterative methods, the resultant eigenvector matrix is orthonormalized, which means as size of the system increases a decreasing magnitude of atomic displacement vector will be experienced. In our visualization module, to avoid this scaling problem, a logistic kernel is applied to fine-tune the atomic magnitude $x_i$ of the eigentensor. First, to avoid eccentricity in the atomic magnitudes, they are clipped between (0.025, 0.975) quantile of the magnitudes. Second, a logistic kernel is applied on the clipped magnitude $x_i$ for the i-th atom

$$y_i = G(x_i|\vartheta) = \frac{1}{(1 + \vartheta e^{-x_i})} < 1 \tag{31}$$

where $\vartheta = 0.05$. To remove the effect of scaling on the normalized eigenvector as the number of atom increase, the output magnitudes are further centered at $\widetilde{y}_i = : \frac{y_i - y^-}{y^+ - y^-} \leq 1$, where $(y^-, y^+)$ are the minimum and maximum of $y_i \forall i$.

**Reporting summary**

Further information on research design is available in the Nature Portfolio Reporting Summary linked to this article.

## Data availability

The data generated in this study, including the atomic coordinates and benchmarks, have been deposited in the Zenodo database under accession code 8087817[89]. Source data are provided with this paper.

## Code availability

The INCHING source code is available on GitHub[90] https://github.com/jhmlam/Inching. Jupyter notebooks and Python scripts for the experiments and analyses presented in the paper are available. Frozen versions of the software and associated code for analysis are also available in the Zenodo database under accession code 10645601[91]. All source codes are provided under an Apache License 2.0 license.

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

## Acknowledgements

This study was supported by internal funding from USC Dornsife college to V.K., and an NSF grant, OAC-2118061 to A.N. The authors acknowledge the Center for Advanced Research Computing (CARC) at the University of Southern California for providing computing resources that have contributed to the research results reported within this publication. URL: https://carc.usc.edu.

## Author contributions

J.H.L., A.N. and V.K. conceived the research. A.N. and V.K. supervised the research. J.H.L. developed the algorithms and led the project under the guidance of A.N. and V.K. All authors discussed the results and contributed to the writing of the manuscript.

## Competing interests

The authors declare no competing interests.
