## [Peer Review File · Nature Communications]

Scalable computation of anisotropic vibrations for large macromolecular assembliesREVIEWER COMMENTS

Reviewer #1 (Remarks to the Author):

The Normal Mode Analysis (NMA) represents a state-of-the-art methodology employed for the examination of macromolecular vibrations and the elucidation of structural alterations therein. Nonetheless, the analysis of vibrations in exceedingly vast macromolecules via NMA presents a big challenge, primarily due to the substantial computational overhead associated with solving an underpinning eigensystem through matrix diagonalization. .

This manuscript details a new algorithm to efficiently calculate vibrational modes using GPUs, allowing for 250 times faster analysis of one of the largest known macromolecules (2.4 million atoms) while implementing various computational techniques to enhance speed and accuracy.

The algorithm's principal innovations revolve around two pivotal aspects:

- 1) Exploiting the sparsity of the NMA matrix eigenproblem while concurrently preserving a localized dense structure.
- 2) Efficient adaptation of the fundamental algorithmic stage to GPU hardware to minimise time to solution.

The methodology presented within this manuscript is detailed and well substantiated from a scientific perspective. Although analogous concepts have been employed in divergent application domains, the pioneering aspect of this endeavor lies in its tailored adaptation to the realm of NMA.

From an algorithmic standpoint, it is discernible that the manuscript does not introduce groundbreaking concepts, primarily relying on the amalgamation of pre-existing linear algebra implementations and CUDA libraries to yield the final INCHING algorithm. Nevertheless, the manuscript's paramount strength lies in its intelligent application of established techniques to a domain hitherto unexplored.

Upon meticulous scrutiny, my evaluation is that, despite the absence of novelty in the foundational components underpinning the INCHING algorithm, this work indisputably broadens the horizons of NMA by extending its sphere of applicability to significantly larger molecular systems. This noteworthy achievement substantiates its suitability for publication in Nature Communications.

Reviewer #2 (Remarks to the Author):

The authors of this work proposed to compute the vibration modes in the Normal Mode Analysis (NMA) of large biomolecules through GPU computing, by transforming the related sparse eigenvalue decomposition problem to a globally-sparse-yet-locally-dense computation, allowing batched tensor products to be most efficiently executed on GPU level-structure bandwidth-reducing algorithms. The method allows accurate calculation of the first 64 vibrational modes of the largest structure in PDB 3Z22 (2.4 million atoms) with a speedup of two magnitudes.

Overall, the proposed work is useful in applications involving large-scale computations of normal mode analysis. However, I have a number of concerns regarding some technical details and in particular, how to position the novelty and performance of the proposed method with regard to existing work in the high-performance-computing community.

1. Can the authors provide references on "construction and diagonalization of the Hessian is notoriously resistant to parallelization"?
2. "In our Method, we demonstrate that by limiting the permutation to atom ordering, we can always generate a graph isomorph of the original elastic network" - I am a bit confused by this statement. A graph (or its Laplacian) will not change under a re-ordering of the nodes, and so any permutation of the Laplacian matrix will remain isomorphic to the original one, isn't this true?
3. Is there any recent attempt to solve the sparse eigenvalue problem with GPU? The related papers about GPU-based sparse eigensolvers that have been referenced in this work look very limited. The cited papers are either on dense matrices (46) or very old methods in the 90's (31,32,43,44). But there seems to be some very recent work along this direction in the HPC community. It would be helpful if the authors could extend the discussions to recent progresses along this line.
4. The step of permuting the rows/columns to transform the Laplacian matrix to a globally-sparse-yet-locally-dense computation can be achieved to reduce the bandwidth of the matrix. It seems that you need to identify close neighbors of each atom based on the given coordinates of all the atoms in order to perform the permutation. Can you clarify the computational complexity of the 3D-tree version of the RCM algorithm?
5. The authors have shown a speed-up of 250-370 times against baselines. Is this baseline method a sparse eigensolver on GPUs, or just a traditional dense solver on GPU without using any GPU parallelization?
6. The authors considered two steps in NMA, i.e., construction of the Hessian, and the sparse eigensolver. I am trying to clarify the novelty and contributions of each step. In the first step, a permutation algorithm is used based on combining 3D-tree structure with the RCM algorithm; in the second step, the authors re-implemented existing eigen-solvers on GPUs based on the permuted

Laplacian matrix with low bandwidth, with batched tensor product computation on GPU - is it statement appropriate? It would be helpful if the authors could elucidate whether their contributions are primarily due to new algorithms or engineering tricks used in the implementation; if it is a mixture of both, what are their respective weights?

Reviewer #3 (Remarks to the Author):

Summary:

=====

This paper presents an interesting algorithm for computing some of the normal modes of a biomolecule whose energy is described by an elastic potential.

The algorithm is specialized to computing on a Graphics Processor Unit (GPU), allowing for fast computation and application to very large systems.

Different eigensolvers have been tested.

Criticisms:

=====

Advances in structural biology makes it possible to study large biological structures, such as full viral capsid. In parallel, methods are now proposed to perform dynamics analyses on those structures.

The paper by Lam et al proposes an implementation of normal mode analyses on the GPU as one of those methods. While their algorithm has clear advantages, there are questions that need to be addressed:

- 1) A key element of their method is the bandwidth reduction of the Hessian matrix. They use their own version of the Reverse Cuthill McKee (RCM) algorithm to perform this reduction. How do their reduction method compare to other parallel RCM implementations, such as speculative RCM (<https://ieeexplore.ieee.org/document/9460553>) ?
- 2) I assume that the bandwidth of the Hessian is strongly dependent on the cutoff method that defines the pairs of atoms included in the elastic potential. The authors rely on a distance cutoff R_c : how does their method scale with respect to R_c (say from $R_c=6$ to $R_c=14$)?
- 3) The authors use several types of eigensolvers in their implementations, such as Lanczos methods and Jacobi-Davidson methods. Why didn't they use in addition a filtering technique, such as the Chebishev filtering that was used by Koehl (as mentioned in the introduction, page 4)?
- 4) Most examples presented in the paper relate to computing up to 100 normal modes. How would the method perform if say 5000 normal modes were computed? Would the performance be linear as a function of the number of normal modes?

Minor:

- 5) It seems that the largest structure in the PDB is the structure of the faustovirus capsid, PDB code 5j7v, with a little over 26 million atoms. How would the program behave on this large structure? What would be the memory footprint?

6) This is more of a philosophical question: I assume that the hyperthreaded algorithm of Koehl would work on any multicore computer and as such on any desktop computer with a cost below \$3000 dollars. The results presented in the paper refers to an NVIDIA A100 GPU card; such an equipment brings the cost of the corresponding computer to above \$20000. Is the gain in computing time worth this difference?

We would like to thank the Referees for their careful reading of the manuscript and useful suggestions. The manuscript was revised accordingly, addressing all the points raised by the reviewers. This includes additional benchmarks, performed at the reviewers' request, supporting our conclusions.

To facilitate navigation of the revision we provide a brief summary of additional benchmarks and major improvements implemented, with a reference to the comments from reviewers:

- Benchmark in the cutoff radius $R_c=6$ angstrom and $R_c=14$ angstrom (Reviewer #3 Q2 on scaling in cutoff radius, Extended Data Figure 3, 4, 8, Result p.10)
- Implementation of Chebyshev-Davidson Method (Reviewer #3 Q3 on comparison of Chebyshev-accelerated methods, Result p.10, Extended Data Figure 8, 9, 10b-c and Method p. 27-28)
- Implementation of low-pass and band-pass Chebyshev-filtered Thick Restart Lanczos Method (Reviewer #3 Q4 on scaling in number of modes, Result p.10)
- Benchmark in calculating 1000 modes (Reviewer #3 Q4 Extended Data Figure 9-10, Result p.10)
- NMA calculation of a coarse-grained Faustovirus capsid with 5 million pseudo-atoms $R_c=14$ angstrom extracted from the 26 million atom complex (PDBID: 5J7V with 15.5 angstrom resolution). (Reviewer #3 Q5, Extended Data Figure 10a, Result p. 11)
- Discussion on recent trends in GPU-accelerated eigensolvers. (Reviewer #2 Q3 on recent work on GPU-accelerated eigensolvers, Discussion p. 13)

Listed below in blue are point-by-point responses to the Reviewers' comments. The corresponding revisions in the manuscript are highlighted in yellow.

Reviewer #1 (Remarks to the Author):

The Normal Mode Analysis (NMA) represents a state-of-the-art methodology employed for the examination of macromolecular vibrations and the elucidation of structural alterations therein. Nonetheless, the analysis of vibrations in exceedingly vast macromolecules via NMA presents a big challenge, primarily due to the substantial computational overhead associated with solving an underpinning eigensystem through matrix diagonalization.

This manuscript details a new algorithm to efficiently calculate vibrational modes using GPUs, allowing for 250 times faster analysis of one of the largest known macromolecules (2.4 million atoms) while implementing various computational techniques to enhance speed and accuracy.

The algorithm's principal innovations revolve around two pivotal aspects:

- 1) Exploiting the sparsity of the NMA matrix eigenproblem while concurrently preserving a localized dense structure.
- 2) Efficient adaptation of the fundamental algorithmic stage to GPU hardware to minimise time to solution.

The methodology presented within this manuscript is detailed and well substantiated from a scientific perspective. Although analogous concepts have been employed in divergent application domains, the pioneering aspect of this endeavor lies in its tailored adaptation to the realm of NMA.

From an algorithmic standpoint, it is discernible that the manuscript does not introduce groundbreaking concepts, primarily relying on the amalgamation of pre-existing linear algebra implementations and CUDA libraries to yield the final INCHING algorithm. Nevertheless, the manuscript's paramount strength lies in its intelligent application of established techniques to a domain hitherto unexplored.

Upon meticulous scrutiny, my evaluation is that, despite the absence of novelty in the foundational components underpinning the INCHING algorithm, this work indisputably broadens the horizons of NMA by extending its sphere of applicability to significantly larger molecular systems. This noteworthy achievement substantiates its suitability for publication in Nature Communications.

We thank the reviewer for this comment, highlighting the technical rigor and practical impact of the study to “broaden the horizons of NMA”. Although we agree that the study does not introduce groundbreaking concepts, the study overcomes several important limitations.

First, while conventional bandwidth reduction algorithms, which take a fully realized matrix as input, are useful when the matrix is historically preserved and only new offline analyses of it are required, they are not applicable in NMA as the realization of a Hessian matrix with large bandwidth (i.e. the input) is highly inefficient. This is why we tightly integrate the Reverse Cuthill-McKee (RCM) algorithm with a data structure tailored to our problem to break apart this chicken-egg dependency.

Second, beyond scaling up in terms of system size, during the revision, we also explored the computational feasibility of extracting large amount of modes (e.g. 1000) using a highly restricted size of basis set (e.g. 128) to maintain a constant consumption of memory. This was achieved by implementing the spectrum slicing approach devised in Li R. et al. 2016.. *SIAM J. Sci. Comput.* 38, 4 (January 2016), A2512–A2534. <https://doi.org/10.1137/15M1054493>. This approach enabled us to successfully compute the first 1000 modes of the HIV virus capsid, which comprises a staggering 2.4 million atoms, in 63 hours, using a basis set of only 128 vectors. Note that, without the

spectrum slicing approach, this calculation will not be possible as a 1000-mode calculation will require >100GB GPU memory.

To summarize, several novel and substantially modified algorithms implemented in this study are:

- 3DRCM – major modification of RCM to efficiently handle 3D coordinate data and avoid premature realization of the Hessian.
- Optimized Implementation of several advanced eigensolvers on GPU, which required custom kernel programming, optimization and modularized integration of techniques in handling our eigensystem.
- Optimized Implementation of low- and band-pass Chebyshev filters into eigensolvers to reduce memory requirements ((in this revision))

The enhanced method and much improved result is now described in the manuscript on Method p. 27-28, Result p. 10 and Extended Data Figure 8-10. We kindly request your attention to this specific aspect of our work as it demonstrates our commitment to both scalability and efficiency.

Reviewer #2 (Remarks to the Author):

The authors of this work proposed to compute the vibration modes in the Normal Mode Analysis (NMA) of large biomolecules through GPU computing, by transforming the related sparse eigenvalue decomposition problem to a globally-sparse-yet-locally-dense computation, allowing batched tensor products to be most efficiently executed on GPU level-structure bandwidth-reducing algorithms. The method allows accurate calculation of the first 64 vibrational modes of the largest structure in PDB (2.4 million atoms) with a speedup of two magnitudes.

Overall, the proposed work is useful in applications involving large-scale computations of normal mode analysis. However, I have a number of concerns regarding some technical details and in particular, how to position the novelty and performance of the proposed method with regard to existing work in the high-performance-computing community.

Thank you for emphasizing the utility of our algorithm in expanding capabilities of NMA.

A summary of additional benchmarks and improvements implemented in revision is listed in the beginning of this letter, with more details provided below.

1. Can the authors provide references on "construction and diagonalization of the Hessian is notoriously resistant to parallelization"?

Thank you for the opportunity to clarify this point. The notorious nature of the Hessian

matrix construction has been discussed in the work of Koehl (Koehl P. J Chem Theory Comput. 2018 Jul 10;14(7):3903-3919. doi: 10.1021/acs.jctc.8b00338.). However, the explicit indexing approach proposed there is not efficient on GPU which generally favors calculations in dense format. Please also kindly refer to the related question 4 where we explained the motivation.

The notorious nature of the diagonalization of the Hessian can be seen in that practical divide-conquer-recombine approach has been realized only for tridiagonal matrices, otherwise, operations requiring matrix-vector multiplication can be done in an embarrassingly parallelized manner. On the diagonalization of matrices, there are also recent studies (Li R. et al. SIAM J. Sci. Comput. 38, 4 (January 2016), A2512–A2534. <https://doi.org/10.1137/15M1054493>) which attempt to solve the eigen-spectrum in separate intervals. In the revision, we also explored feasibility of this method in obtaining interior eigenpairs allowing us to overcome memory limitations and obtain the first 1000 modes of all benchmarks. Please kindly refer to the related question 3 and Extended Data Figure 9 and Extended Data Figure 10. The references above are added to the manuscript.

2. "In our Method, we demonstrate that by limiting the permutation to atom ordering, we can always generate a graph isomorph of the original elastic network" - I am a bit confused by this statement. A graph (or its Laplacian) will not change under a re-ordering of the nodes, and so any permutation of the Laplacian matrix will remain isomorphic to the original one, isn't this true?

We thank the reviewer for the opportunity to clarify this to the reader. It is true that any permutation of node ordering on the symmetric Hessian network will not change the connectivity i.e. the resultant graph is always isomorphic to the original one. However, the permutation is not trivial as we want atoms grouped together to achieve dense calculations on GPU to fully exploit its parallelization ability. Specifically, while the construction and diagonalization of NMA are regarding the Hessian matrix, we choose only to permute the atom ordering rather than all $3N$ variables. This choice allows us to surrogate a premature realization of both the Hessian matrix and the Laplacian matrix before deciding how to group the atoms together in our RCM algorithm. (Please kindly refer to the related Q4 answer for more details on the motivation.)

We edited p. 6 line 129-137 to clarify on this.

3. Is there any recent attempt to solve the sparse eigenvalue problem with GPU? The related papers about GPU-based sparse eigensolvers that have been referenced in this work look very limited. The cited papers are either on dense matrices (46) or very old methods in the 90's (31,32,43,44). But there seems to be some very recent work along this direction in the HPC community. It would be helpful if the authors could extend the discussions to recent progresses along this line.

We thank the reviewer for suggesting a more extensive discussion of recent studies. GPU-accelerated approaches for solving large sparse symmetric eigenproblems remain a vibrant area of research. However, most recent developments on GPU eigen-solvers are focused on dense matrices. We found a few recent works with sparse routines, including:

- MAGMA (H. Anzt, W. Sawyer, S. Tomov, P. Luszczek, I. Yamazaki and J. Dongarra, 2014 IEEE International Parallel & Distributed Processing Symposium Workshops, Phoenix, AZ, USA, 2014, pp. 941-949, doi: 10.1109/IPDPSW.2014.107.)

While the focus of the library is on dense routine, a Locally Optimal Block Preconditioned Conjugate Gradient (LOBPCG) sparse routine is provided. https://icl.utk.edu/projectsfiles/magma/doxygen/group_magmasparse_dsyev.html The LOBPCG requires to compute a preconditioner matrix, through for example an incomplete LU factorization, but the fill-in level of the preconditioner will further increase memory burden. This may be applicable to scenarios where medium-sized Hessian matrix (and corresponding preconditioner) is of interest, but in our case we are dealing with very large sparse Hessian matrices.

- Top-K GPU eigen-solver (F. Sgherzi, A. Parravicini and M. D. Santambrogio, 2022 IEEE International Symposium on Circuits and Systems (ISCAS), Austin, TX, USA, 2022, pp. 1259-1263, doi: 10.1109/ISCAS48785.2022.9937893)

In this work, the authors developed a multi-GPU, mixed precision, Top-K eigensolver, to resolve the largest eigenvalues in a sparse matrix. The work will be very useful in domain such as spectral clustering, while in our case, we are interested mostly in the smallest eigenvalues.

- ChASE (Xinzhe Wu, Davor Davidović, Sebastian Achilles, and Edoardo Di Napoli, . In Platform for Advanced Scientific Computing Conference (PASC '22), June 27–29, 2022, Basel, Switzerland. ACM, New York, NY, USA, 12 pages. <https://doi.org/10.1145/3539781.3539792>)

The method is based on Chebyshev polynomial filtered subspace iteration. In the publication, they also mentioned that the matrix can be dense, sparse or banded.

The discussion in p. 13 was updated with these studies and all the above references were added in the manuscript.

4. The step of permuting the rows/columns to transform the Laplacian matrix to a globally-sparse-yet-locally-dense computation can be achieved to reduce the bandwidth

of the matrix. It seems that you need to identify close neighbors of each atom based on the given coordinates of all the atoms in order to perform the permutation. Can you clarify the computational complexity of the 3D-tree version of the RCM algorithm?

We thank the reviewer for this question. In the Method section p. 20-22, we showed that both the 3DRCM and the original Reverse Cuthill-McKee (RCM) algorithm are both dominated by a pseudolinear time complexity, and the neighbors are dynamically retrieved and erased; it is not a computationally expensive step. However, there is an important distinction between our 3DRCM algorithm with other RCM algorithms. Namely, their input is a matrix already realized, i.e. the input matrix is already given and stored, which may be useful when the matrix is historically preserved and only new offline analyses of it are required. A major problem overlooked is that in many scenarios, it is not even possible to realize the matrix efficiently in parallel when they span large bandwidths. Without the RCM order, we cannot efficiently compute the matrix; without the matrix, we cannot run conventional RCMs which take matrices as input. This is also why we attempted to tightly integrate the RCM with a data structure tailored to our problem to break apart this chicken-egg dependency.

In Extended Data Figure 3 and 4, we now included a new benchmark on different cutoff radii $R_c = 6$ and 14 angstroms to show that the 3DRCM behaves consistently when there are smaller or larger number of neighbors. Note that for a larger $R_c = 14$ angstrom, 64-bit indexing instead of 32-bit was used throughout to accommodate larger amount of non-zero elements in the matrix. Corresponding NMA calculation on these new benchmarks are presented in Extended Data Figure 8.

These results are now included in p. 10 line 226-228

5. The authors have shown a speed-up of 250-370 times against baselines. Is this baseline method a sparse eigensolver on GPUs, or just a traditional dense solver on GPU without using any GPU parallelization?

We take baseline method as CPU sparse eigensolver implemented on ARPACK as this is a reliable implementation as well as a popular choice in similar programs such as ProDy and it was also compared in the work of Koehl (Koehl P. J Chem Theory Comput. 2018 Jul 10;14(7):3903-3919. doi: 10.1021/acs.jctc.8b00338.). We did not compare with dense GPU eigensolver that may possibly require more than one GPUs, but the GPU definitely cannot hold the dense Hessian matrix in memory in double precision when there are $> 100k$ atoms, $((3 \times 100000)^2) \times 8 \text{ byte} = 670\text{GB}$. For a comparison with a sparse eigensolver on a single GPU, we compared with CuPy's default implementation of the Thick Restart Lanczos Method, which we optimized on its performance as it failed to accommodate larger systems (Method p. 29-30).

Beyond scaling up in terms of system size, during the revision, by incorporating the band-pass Chebyshev filter in Li R. et al. SIAM J. Sci. Comput. 38, 4 (January 2016), A2512–A2534. <https://doi.org/10.1137/15M1054493>, we also explored the

computational feasibility to scale up in terms of the number of modes while avoiding excessive GPU memory usage. (See Extended Data Figure 9 and 10) This allows us to calculate the first 1000 modes of the 2.4 million atom system without compromising memory in linear run time. In addition, methods with low-pass filter were also compared in Extended Data Figure 8 and Method p.27-28.

We kindly request your attention to this specific aspect of our work as it demonstrates our commitment to both scalability and efficiency.

6. The authors considered two steps in NMA, i.e., construction of the Hessian, and the sparse eigensolver. I am trying to clarify the novelty and contributions of each step. In the first step, a permutation algorithm is used based on combining 3D-tree structure with the RCM algorithm; in the second step, the authors re-implemented existing eigensolvers on GPUs based on the permuted Laplacian matrix with low bandwidth, with batched tensor product computation on GPU - is it statement appropriate? It would be helpful if the authors could elucidate whether their contributions are primarily due to new algorithms or engineering tricks used in the implementation; if it is a mixture of both, what are their respective weights?

We thank the reviewer for this question, that allows us to clarify the novelty aspects of our approach. In the first Hessian construction step, we developed a new algorithm by incorporating a 3D-tree data structure into the RCM approach. Conventionally RCM requires the whole matrix as an input, which is unrealistic because the computation of their input already requires efficient dense tensor product parallelization on GPU. This tight integration of the RCM with consideration of data structure before the realization of the matrix itself is novel.

For the second part on eigen-solver, we are not proposing a new algorithm to solve eigenproblem, which is a vibrant field of development. However, careful tailoring and integration of new techniques with mathematical understanding is required to produce fast accurate result as most of these algorithms were written as templates targeting broad classes of matrices. Without these “engineering tricks”, calculation will be impossible.

Both are non-trivial aspects of our work leading to previously unexperienced results. We would say, it is a mixture of new algorithms and “engineering tricks” and at least 50% is due to new algorithms.

To summarize, several novel and substantially modified algorithms implemented in this study are:

- 3DRCM – major modification of RCM to efficiently handle 3D coordinate data and avoid premature realization of the Hessian.
- Optimized Implementation of several advanced eigensolvers on GPU, which required custom kernel programming, optimization and modularized integration of techniques in handling our eigensystem.

- Optimized Implementation of low- and band-pass Chebyshev filters into eigensolvers to reduce memory requirements ((in this revision))

Reviewer #3 (Remarks to the Author):

Summary:

=====

This paper presents an interesting algorithm for computing some of the normal modes of a biomolecule whose energy is described by an elastic potential. The algorithm is specialized to computing on a Graphics Processor Unit (GPU), allowing for fast computation and application to very large systems. Different eigensolvers have been tested.

We thank the reviewer for carefully reading our work.

A summary of additional benchmarks and improvements implemented in revision is listed in the beginning of this letter, with more details provided below.

Criticisms:

=====

Advances in structural biology makes it possible to study large biological structures, such as full viral capsid. In parallel, methods are now proposed to perform dynamics analyses on those structures. The paper by Lam et al proposes an implementation of normal mode analyses on the GPU as one of those methods. While their algorithm has clear advantages, there are questions that need to be addressed:

1) A key element of their method is the bandwidth reduction of the Hessian matrix. They use their own version of the Reverse Cuthill McKee (RCM) algorithm to perform this reduction. How do their reduction method compare to other parallel RCM implementations, such as speculative RCM (<https://ieeexplore.ieee.org/document/9460553>) ?

We thank the reviewer for this question. There is an important distinction of our RCM algorithm with other RCM algorithms, including the speculative RCM, namely, their input is a matrix already realized, i.e. the input matrix is already given and stored, this may be useful when the matrix is historically preserved and only new offline analyses of it are required. However, in our case, the production of this unpermuted matrix already assumes parallel computation that possibly spans large bandwidths. Without the RCM order, we cannot efficiently compute the matrix; without the matrix, we cannot run any

RCMs which take matrices as input. This is why we tightly integrate the RCM with a data structure tailored to our problem to break apart this chicken-egg dependency.

This motivation is now clarified in p. 6 line 129-138. The speculative RCM is also cited.

2) I assume that the bandwidth of the Hessian is strongly dependent on the cutoff method that defines the pairs of atoms included in the elastic potential. The authors rely on a distance cutoff R_c : how does their method scale with respect to R_c (say from $R_c=6$ to $R_c=14$)?

We thank the reviewer for this question. The 8 angstrom R_c is taken from Koehl (Koehl P. J Chem Theory Comput. 2018 Jul 10;14(7):3903-3919. doi: 10.1021/acs.jctc.8b00338.) and Tirion (Tirion MM. Phys Rev Lett. 1996 Aug 26;77(9):1905-1908. doi: 10.1103/PhysRevLett.77.1905.) for consistency.

We now also provide the benchmark with R_c in 6 angstroms and 14 angstroms. It is true that the bandwidth of the Hessian depends on the distance cutoff size. However, in Extended Data Figure 3, we show that the reordered bandwidth is still consistently better than the original in all cases. Run time of 3DRCM and the computation of Hessian were also consistently linear in all cases. In Extended Data Figure 8 d, we showed that linear or better scaling in overall run time has been achieved in terms of the radii of interactions to be considered, except several cases at a lower radius of interaction at 6Å, likely due to poorer conditioning of the matrix. The memory growth is also consistent with the number of neighbors encountered in the Hessian (Extended Data Figure 4 and 8 e) Note that comparison of scaling in radii in memory is dependent on the number and indexing of non-zero entries in Hessian, where for all 14 angstrom calculations, 64-bit indexing were used instead of 32-bit indexing to accommodate the growth in non-zero entries.

These results are now included in p. 10 line 226-228

3) The authors use several types of eigensolvers in their implementations, such as Lanczos methods and Jacobi-Davidson methods. Why didn't they use in addition a filtering technique, such as the Chebishev filtering that was used by Koehl (as mentioned in the introduction, page 4)?

We thank the reviewer for this question. For the initial submission, we focus on methods without hyperparameters, a major one is the degree of the Chebishev polynomial. The Koehl 2018 study shows that, for a moving low-pass first-kind Chebishev filter, the optimal value can vary in range of 5 to more than 80 depending on the diagonalization method and the atomic system. Indeed, the optimal degree for the extremal interval containing first k eigenvalues will depend on the spectral density of individual matrices.

In the revision, we now also incorporated the Chebyshev-Davidson method (INCHING-CDM) and a Thick Restart Lanczos Method with low-pass Chebyshev filters (INCHING-CTRLM) into our protocols. The optimal polynomial degree is now determined by maintaining the convergence rate a constant in the transformed eigenproblem (See Method p. 27-28 and Extended Data Figure 10b for illustration). In Extended Data Figure 8, we show that, at the same accuracy level as JDM (10^{-12}), moderate speed-up in the sub-megascale regime (mean at 1.21) can be achieved by CDM with an 80-degree polynomial, though the speed-up diminished to a slow-down in the megascale regime (mean at 0.91). This contrasts with applying the optimized static low-pass filter to TRLM, where we showed that the Chebyshev-filtered TRLM (CTRLM) can steadily deliver a speed-up over TRLM in the megascale regime (mean at 7.44) and is faster than CDM and JDM.

In the revision, we also explored the feasibility of a band-pass Chebyshev filter (polynomial expansion on a Dirac-delta-like function damped by a Jackson kernel) coupled with CTRLM to overcome memory limit in approaching the first 1000 normal modes. Please kindly refer to our answer in below Question 4. (Extended Data Figure 9 and 10)

These results are now included in p. 10 line 223

4) Most examples presented in the paper relate to computing up to 100 normal modes. How would the method perform if say 5000 normal modes were computed? Would the performance be linear as a function of the number of normal modes?

We thank the reviewer for this question, prompting us to explore new approaches to overcome the challenge of expanding the number of modes. The low-pass Chebyshev filtered methods e.g. Chebyshev Davidson (CDM) as in the work of Koehl (Koehl P. J Chem Theory Comput. 2018 Jul 10;14(7):3903-3919. doi: 10.1021/acs.jctc.8b00338) were designed for solving a limited number of extremal eigenvectors, where converged eigenvectors are required to be accessed for orthogonalization. As a result, the 5000-modes computation in Koehl were achieved with a 71k coarsened atomic systems, requiring 16GB RAM just to store the total workspace of the Ritz vectors, which is often at least 2 times of the number of wanted eigenvectors. This memory requirement scales linearly with the system's number of atom. For example, a 710k atom system for 5000 eigenvectors will require 160GB; a 2.4 million system for a 1000 mode calculation will require 107 GB, far exceeding that of our GPU hardware. While the focus of our submitted manuscript was not on the higher modes for megascale system as we did not find strong applications, it does illustrate infeasibility of similar diagonalization methods to scale in terms of modes, including CDM and those we tested.

Importantly, during the revision, we found a way to overcome this limit. This is achieved by exploring the use of a band-pass filter derived in Li R. et al. 2016. SIAM J. Sci. Comput. 38, 4 (January 2016), A2512–A2534. <https://doi.org/10.1137/15M1054493>, which allows

interior eigenpairs to be solved. The band-pass filter we referred to is a Chebyshev expansion of a Dirac-delta-like function damped by a Jackson kernel (c.f. the low-pass filter as implemented in CDM, see Extended Data Figure 10b). In Extended Data Figure 9, we showed that, by coupling this band-pass Chebyshev filter with our TRLM implementation, linear time scaling in the number modes was achieved while maintaining GPU memory at constant of a very limited size of basis set. This ultimately allows the first 1000 modes of all the benchmarks, including a 2.4 million atom system, to be solved in 63 hours maximum without compromising memory or run time. Note that for consistency, a highly limited workspace of 128 vectors was adopted for all systems, though smaller system can afford much larger workspace. In Extended Data Figure 10c, we also demonstrate that this strategy works on a cheap gaming GPU (see related Q6 answer).

The implementation and results are now described in the manuscript on p. 10 line 223 and Method p. 27-28

Minor:

5) It seems that the largest structure in the PDB is the structure of the faustovirus capsid, PDB code 5j7v, with a little over 26 million atoms. How would the program behave on this large structure? What would be the memory footprint?

We omitted this structure from our benchmarks due to its low resolution (15.5Å). However, we admit that a 26 million-atoms structure is a challenge to our algorithms optimized for single GPU due to limited on-board memory of current GPU chips (~80Gb). Therefore, we cannot store the >178GB Hessian triangle required for the 26 million atom system, nor afford to sacrifice run time using an on-the-fly tensor product approach, even though the time to calculate all segments of the matrix without storage is estimated to take less than 1.2 hour. The 3DRCM memory footprint is acceptable (<2GB) for the un-coarsened structure. Although both low resolution and memory limitations for the current hardware are not conducive to all-atom NMA calculations, we now also include NMA calculation of a coarse-grained Faustovirus capsid system with 5 million pseudo-atoms with $R_c = 14$ angstrom. (Extended Data Figure 10a).

The result of the coarse-grained system is now included in p. 11 line 260-265 and the limitation is mentioned in p.13 line 310-312

6) This is more of a philosophical question: I assume that the hyperthreaded algorithm of Koehl would work on any multicore computer and as such on any desktop computer with a cost below \$3000 dollars. The results presented in the paper refers to an NVIDIA A100 GPU card; such an equipment brings the cost of the corresponding computer to above \$20000. Is the gain in computing time worth this difference?

We thank the reviewer on this practical philosophical question which we all face in our research. We now tested that the INCHING algorithm also works on our lab desktop equipped with a \$2500 RTX 4090 with only 24GB memory for the 2.4 million atoms HIV

capsid. (Extended Data Figure 10c) With the band-pass filter introduced in the revision (See Question 4), the memory limit regarding Ritz vectors were alleviated. This allows us to resolve the first 200 mode of this 2.4 million atoms system in batches of 28 eigenvectors, in less than 27 hours; larger batch size is also affordable with 32-bit indexing. Our INCHING algorithm also works on a low-end T4 GPU provided by Google Colab for free. We hope that these gains in computing time as well as the handle for memory in lower-end equipment show our worth.

The results are now described in the manuscript on p. 10 line 236-237 and Method p. 29 line 688

REVIEWERS' COMMENTS

Reviewer #1 (Remarks to the Author):

The Authors have address all my outstanding concerns, and my assessment is that the manuscript is now suitable for publication in *Nature Communications*.

Reviewer #2 (Remarks to the Author):

The authors of the paper have clarified several points regarding my previous comments, including the novelty of the method and evaluation settings. Overall I believe it is suited for publication in the *Nature Communications*.

Reviewer #3 (Remarks to the Author):

The authors answered all my criticisms. I believe that the revised manuscript is fit for publication in *Nature Communications*.